# communications
# engineering

# Real-time 4D tracking of airborne virus-laden droplets and aerosols

Devendra Pal [1], Marc Amyot [2], Chen Liang[3] & Parisa A. Ariya [1,4]✉

There is currently no real-time airborne virus tracking method, hindering the understanding of rapid virus changes and associated health impacts. Nano-digital in-line holographic microscopy (Nano-DIHM) is a lensless technology that can directly obtain the interference patterns of objects by recording the scattered light information originating from the objects. Here, we provide evidence for real-time physicochemical tracking of virus-laden droplets and aerosols in the air using desktop label-free Nano-DIHM. The virus interference patterns, as single and ensemble particles, were imaged by the Nano-DIHM with 32.5 ms resolution. The next-generation Stingray and Octopus software was used to automate object detection, characterization and classification from the recorded holograms. The detection system was demonstrated to detect active MS2 bacteriophages, inactivated SARS-CoV-2 and RNA fragments, and an MS2 mixture with metallic and organic compounds. This work demonstrates the feasibility of using Nano-DIHM to provide rapid virus detection to improve transmission management in real time.

[1] Department of Atmospheric and Oceanic Sciences, McGill University, 805 Sherbrooke Street West, Montreal, QC H3A 0B9, Canada. [2] Department of Biological Sciences, Univerité de Montréal, Complexe des Sciences, 1375 Avenue Thérèse-Lavoie-Roux, Montréal, QC H2V 0B3, Canada. [3] Department of Medicine, Division of Experimental Medicine, McGill University and Jewish General Hospital, 3755 Cote Sainte Catherine Rd., Montreal, QC G3T 1 E2, Canada. [4] Department of Chemistry, McGill University, 801 Sherbrooke Street West, Montréal, QC H3A 2K6, Canada. ✉email: parisa.ariya@mcgill.ca

A recent study has identified a significant knowledge gap is the in-situ and physicochemical characterization technology of airborne viruses to better model viral transmission[1,2]. Until now, no in situ and real-time technology for physicochemical airborne virus characterization exists[1,2]. The existing technologies are insufficient for detecting a single airborne virus or even clusters of them in the air and cannot be used to decipher transmission mechanisms accurately[1,2]. We present a robust imaging technology addressing the knowledge gap by providing much-needed in-situ and real-time capability and tracking viruses in the air.

The transmission mechanism(s) of SARS-CoV-2 is(are) still debated. However, almost all studies point to the fact that SARS-CoV-2 is transmitted by exposure to infectious respiratory fluids[3–6]. Three major transmission pathways for respiratory diseases are the (a) inhalation of viral droplets and airborne particles or bioaerosols[7,8], (b) deposition of exhaled aerosols and droplets onto exposed mucous membranes[9,10], and (c) physical contact of exhaled viral aerosols and droplets on surfaces[11,12]. The best current non-in-situ technologies, such as real-time polymerase chain reactions, have poor detection limits and require approximately 100 copies of viral RNA per milliliter of transport media[2]. These transmission pathways can take place simultaneously and are multifactorial processes affected by factors such as viral load, duration of contact, concentration[13], multiple exposures, environmental conditions, host receptor age, immune system, etc[14–16]. Numerous research studies worldwide agree that facial masks show a systematic decrease in SARS-CoV-2 transmission, demonstrating the importance of air as a significant transmission pathway[17–19], but little is known about it[1].

Air is in motion, as are airborne virus-laden particles. Airborne viral droplet or aerosol detection and quantification methods have generally included collecting samples into liquids or onto solid surfaces that are not in situ or real-time[20–23]. Various conventional techniques exist to observe, collect, and quantify viruses, such as molecular assays, immunoassays, multistage collectors, fluorescent sensors, and several types of polymerase chain reactions, as shown in Supplementary Information (SI) Table S1. However, despite their advantages, these techniques are neither real-time nor in situ techniques[21–26] and, therefore, cannot provide a physicochemical understanding of dynamic air.

Suspensions of virus-laden droplets in the air are classified as a subset of bioaerosols[27], containing microbiological entities such as bacteria, pollens, fungi, dead or alive viruses, and biological activity markers[28]. During recent decades, it has become clear that bioaerosols can undergo physicochemical transformations[29,30]. Bioaerosols interact with gases and airborne particles, forming complex mixed structures[31,32] that transform under different environmental conditions, such as temperature, humidity, radiation and air dynamics, and at various atmospheric interfaces, such as built surfaces, snow/ice, water, and soil[33]. The lack of physicochemical transformation of airborne viruses precludes scientists from foreseeing even viral aerosol evaporation and condensation processes[1], which are pivotal for primary and secondary viral transmission[1,15].

This study used Nano-Digital in-line Holographic Microscopy (Nano-DIHM) to investigate viruses in air and water in situ in real time. The Nano-DIHM comprises a desktop holographic microscope (4Deep, Halifax, Nova Scotia)[34] and a gas flow tube that allows airborne particles to travel through the imaging volume of the DIHM, enabling real-time observation of single or ensembles of viral particles or other objects. Nano-DIHM is a lensless technology that directly records interference patterns called holograms of the incident and scattered light using a light-sensitive matrix/digital camera[34–36]. The object information was recovered from the recorded holograms by performing numerical reconstruction using Octopus/Stingray software based on a patented algorithm[37,38]. We observed the in situ and real-time physicochemical transformation of the virus-laden droplets and aerosols of two viruses, SARS-CoV-2 and MS2 bacteriophage (MS2), which is widely used as a surrogate for viruses in the air. Using artificial intelligence, nano-DIHM enables 4D tracking of viruses in air and water and has the capability for surface measurement (e.g., roughness and surface areas) in situ and in real time. It does not require particle trapping, collection, or virus particle deposition, and no strong laser is needed. The Methods section provides a schematic, experimental procedure and workflow of nano-DIHM. The experimental parameters for each experiment are shown in Table S2. We used SARS-CoV-2 particles with confirmed whole genome sequencing (Fig. S1). Inactivated SARS-CoV-2 was provided by the Contaminant Level 3 Platform in the Faculty of Medicine at McGill University.

We first provide evidence for the observation of live MS2 virus particles (3D size, phase, intensity, surface properties and their dynamic trajectories), and we present the validation of nano-DIHM results with scanning/transmission electron microscopy (S/TEM) as a single virus particle and an ensemble of them. Second, we demonstrate the nano-DIHM capability of detecting and classifying SARS-CoV-2 particles and their physiochemical properties in dynamic (sneezing-coughing) and static modes (SARS-CoV-2 deposited on a microscope slide). Photochemical experiments are performed to confirm the detection accuracy and physicochemical characteristics of observed viruses even after deactivation. Then, we present a library of individual classifiers for automated software in YES/NO format, which allows SARS-CoV-2 particle identification in dynamic air. In addition, we establish that we can clearly distinguish specific viruses in inorganic mixtures by identifying SARS-CoV-2, SARS-CoV-2 RNA, MS2, and metals such as $TiO_2$ and $Fe_2O_3$ in the mixed samples. We establish that we can selectively detect viruses in organic mixtures (MS2, alpha-pinene, oil, and honey). Finally, we discuss this technology's potential in various research domains, including non-invasive medical imaging.

## Results and discussion

**Validation of the MS2 bacteriophage shape and size obtained by nano-DIHM with S/TEM.** Our first step was to validate the potential of nano-DIHM to detect virus characteristics using a commonly used airborne virus surrogate: MS2. MS2 was purchased from ZeptoMetrix (Buffalo, NY, USA), and detailed information is described in the Methods section. In the current setup, laser light (405 nm) illuminates particles suspended in the air or water as they flow through a flow tube cuvette or are placed on a microscope slide, and holograms of the particles are recorded onto a screen. A fast image-processing software, Octopus/Stingray, based on a patented algorithm[37,38], analyzes the holograms to extract individual dimensions, phases, shapes, and surfaces of individual particles[39]. The virus sizes, phase, morphology, surface area, and roughness in dynamic and stationary media are given in Table S3. The active MS2 virus size, shape, and morphology measurements made by nano-DIHM were validated using high-resolution S/TEM and Talos-S/TEM, as shown in Fig. 1. The airborne MS2 viral particle size distribution determined by nano-DIHM was confirmed with a Scanning Mobility Particle Sizer (SMPS), Optical Particle Sizer (OPS), and Particle Size Analyzer (PSA) (Fig. 2). The electron microscopy and particle sizer details are given in the Methods section.

Observations of MS2 viruses in the aqueous mode were obtained by nano-DIHM and validated with independent results obtained by S/TEM and Talos-S/TEM with the same sample shown in Fig. 1. The intensity reconstruction of the MS2 viruses

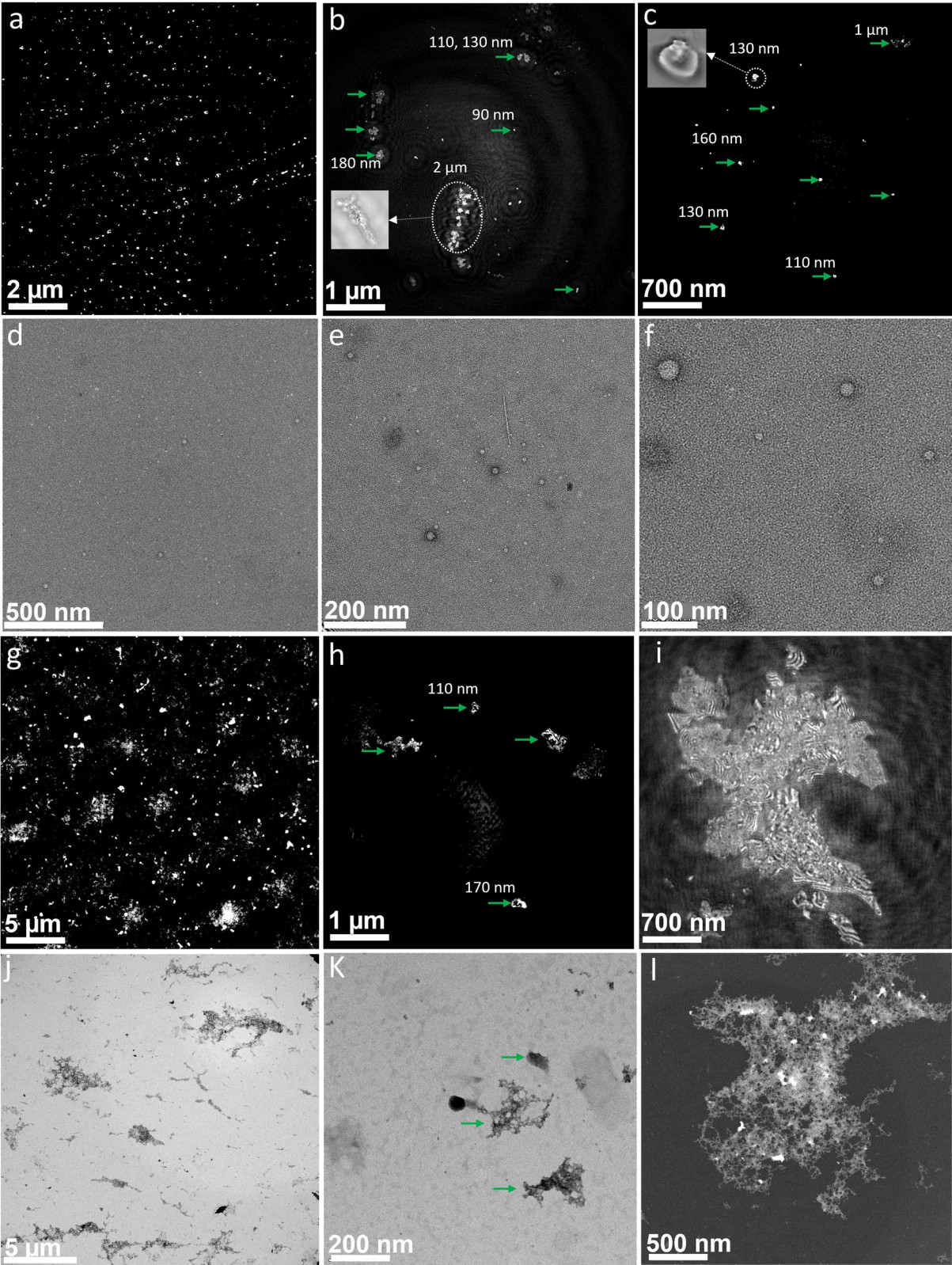

obtained by nano-DIHM is displayed in Fig. 1a–c and d–f presents the MS2 images obtained by Talos-S/TEM. The MS2 virus samples were negatively stained (methods section), as shown in Fig. 1a–f. In Fig. 1b, c, the circles and green arrow indicate MS2 virus sizes of ~ 90 nm, 110 nm, 130 nm, 160 nm, 180 nm, 1 µm, and 2 µm. Fig. 1b and c inset images are the zoom-in images in circles (Fig. 1b, c). We have provided the

information in figure captions The size and shape determined by nano-DIHM (Fig. 1b, c) were validated by the Talos-S/TEM images, as shown in Fig. 1d-f. The intensity and phase profiles across the crosscut of the MS2 particles shown in Fig. 1b, c are shown in Fig. S2a–f. During the reconstruction process, the background holograms recorded in the absence/without any particles were subtracted from the raw holograms recorded in the

**Fig. 1 Size and morphology of MS2 particles obtained by nano-DIHM in a stationary manner and results validated by Talos-S/TEM and S/TEM.** The top panel (**a**–**c**) shows the intensity reconstruction obtained by nano-DIHM. **a** Intensity reconstruction of MS2 viruses at a reconstruction distance of Z = 2844 μm, **b** intensity reconstruction at Z = 1132 μm, with the inserted image in (**b**) showing the magnified MS2 virus shape, and **c** reconstruction performed Z = 702 μm and nanosized MS2 viruses observed in high-resolution images. Zoomed-in images in white dotted circles in (**b**, **c**) show an example of the MS2 virus shape and size. The green arrows indicate that several nano- to microsized MS2 viruses existed at the same reconstruction distance. **d**–**f** The Talos-S/TEM images of MS2 viruses exhibiting the shape and size. The results in (**a**–**e**) were obtained with negatively stained MS2 samples. **g**–**i** Intensity reconstruction of 100X diluted MS2 samples without negative staining reconstructed at reconstruction distances of Z = 3267 μm, Z = 496 μm, and Z = 367 μm using nano-DIHM. **j**–**l** S/TEM images from the same samples. Both the nano-DIHM and S/TEM results confirmed the agglomeration of MS2 viruses without negatively stained samples.

presence of the sample to remove any experimental/optical impurities[39]. More detailed hologram recordings/reconstruction and automatic virus detection and classifications are given in the method section.

To reproduce a more realistic situation of measuring viruses in a natural environment, we diluted the original MS2 samples in Milli-Q water by 100 X volume. Furthermore, we mixed the MS2 viruses with several organics and metal oxides (see below, metal coating section). An example of MS2 virus detection in the 100 X diluted sample by nano-DIHM is shown in Fig. 1g–i, and the corresponding S/TEM images are displayed in Fig. 1j–l. The hydrodynamic size peak of MS2 in water was observed at ~ 200 nm, and 700 nm by PSA (Fig. S2g, h) and was aligned with the size data determined by nano-DIHM (Table 1). Interestingly, the morphology of MS2 in the 100 X diluted samples changed from spherical to aggregate, suggesting that water uptake and/or MS2 coagulation occurred[40,41]. The negatively stained MS2 samples did not show the aggregated morphology because the stained materials created a dark border around the MS2 viruses[42], which is unlikely in the atmosphere. Nano-DIHM not only detected single virus particles but also confirmed the presence of agglomerates/clusters (Fig. 1g–i). The nano-DIHM determined and classified the overall shape of virus particles, but it did not decipher the precise cluster shape of MS2 as compared to the high-resolution S/TEM. Nano-DIHM offers promising results for determining the phase, shape, size, and surface properties of airborne/waterborne particles. Currently, high-resolution electron microscopy is a powerful measurement tool for determining virus shape/morphology[21,31,43]. But, electron microscopy does not have the in situ and real-time imaging capabilities[21,31,43] that nano-DIHM offers[39] on the millisecond time scale.

**Tracking of aerosolized MS2 characteristics in dynamic air.** Nano-DIHM efficiently determined the phase, size, shape, and surface properties of airborne MS2 particles with dimensions ranging from nano- to micrometers in dynamic media (Fig. 2, Table 1). Table 1 provides detailed experimental setup and statistical information on three aerosolization types of MS2 particle size distributions, and Supplementary movies S1 and S2 provide the 4D (time and 3D positions) dynamic trajectories. 1) MS2 samples were aerosolized with a C-flow atomizer. Afterward, the aerosol stream passed through two diffusion dryers to the nano-DIHM sample volume (flow tube cuvette) and outlet connected to the SMPS/OPS (Method section, Schematic Figure). The humidity was <4%. 2) A mixed solution of MS2 and TiO2 was aerosolized, and the aerosol stream passed through a diffusion dryer. 3) A bubbler was used to generate MS2 droplets. The droplets directly passed through a flow tube cuvette during nano-DIHM. No dryer was used.

The aerosolized MS2 viral particle size distribution determined by nano-DIHM varied from the nano- to microscale (Table 1). This result was consistent with the simultaneous analysis performed using the SMPS and OPS in the particle size range

from 10 nm to 10 μm (Fig. S3a–d). The SMPS and OPS only measure particle size distributions in real time[31] and unable tracking virus/particle trajectories in 3D space and individual virus particle dimensions in 3D, in contrast to nano-DIHM[39]. Nano-DIHM is very versatile and can easily be coupled to various particle analyzers, PCR equipment, or a wide range of electronic microscopy units.

The phase and intensity results of airborne MS2 viral particles shown in Fig. 2a–f indicate that the MS2 particles existed in varied shapes and morphologies, from spherical to irregular. It is unlikely that the virus/material would maintain uninterrupted morphologies during aerosolization without attaching to the suspending matrix[44]. The intensity profiles for the particle crosscuts in Fig. 2a–c are illustrated in Fig. 2g–i, while their phase response is shown in Fig. 2j–l. Negative and positive phase shifts across the MS2 particle crosscuts were observed, varying from 2.6 to 4 radians. This could be due to MS2 particle coagulation or aggregation during the aerosolization process[44] or self-interaction among the MS2 particles that enlarged the size of the particles. The MS2 particle sizes determined by the nano-DIHM in Fig. 2a are 80 and 290 nm, and those in Fig. 2b are 130, 80, and 210 nm, while the particles in Fig. 2c are submicron to 2 and 3 μm, expressed as the full width at half maximum (FWHM) (Fig. 2g–i).

The 3D orientation (XYZ positions) and individual dimensions (width, height, and length) of airborne MS2 viral particles in a single hologram with a temporal resolution of 31.25 ms in moving air are shown in Fig. S6. The detailed statistics of airborne MS2 particles (dry aerosols), MS2 samples mixed with TiO2 (dry aerosols), and MS2 particles in sneezed/droplets form in 3D space are given in Table 1. The median values of width, height, and length of the airborne MS2 particles/vesicles (dry aerosols) determined by nano-DIHM are 180 nm, 180 nm, and 380 nm, respectively. The median size distributions of a mixed suspension of MS2 with TiO2 are 340 nm, 330 nm, and 660 nm, respectively. The median size distribution of MS2 viruses in the droplet (sneezing-coughing) form increased toward a more prominent size, and values were observed at 770 nm, 730 nm, and 1.12 μm, respectively. This may be due to the moist envelope across the MS2 virus vesicles.

Nano-DIHM and SMPS/OPS observations indicated that the airborne MS2 viral particle (dry aerosol) size peaks were between 60 and 200 nm. The size distribution of MS2 particles in the TiO2 suspension is shown in Fig. S3c, d. The mixed TiO2 and MS2 suspension exhibited a shifted size distribution to 150–350 nm. This could be due to attachment or coagulation/aggregation between MS2 and TiO2 particles[40,41]. The high-resolution electron microscopy images in Figs. 6 and S9 confirmed the attachment of TiO2 on MS2 viral particles. The mixing/coating of TiO2 in MS2 samples suggested physicochemical transformation and is essential to consider for the secondary transmission of airborne viruses. This interesting phenomenon between MS2 (as a surrogate of virus or viral analog) and TiO2 (abundant cosmetic material) indicates the physicochemical

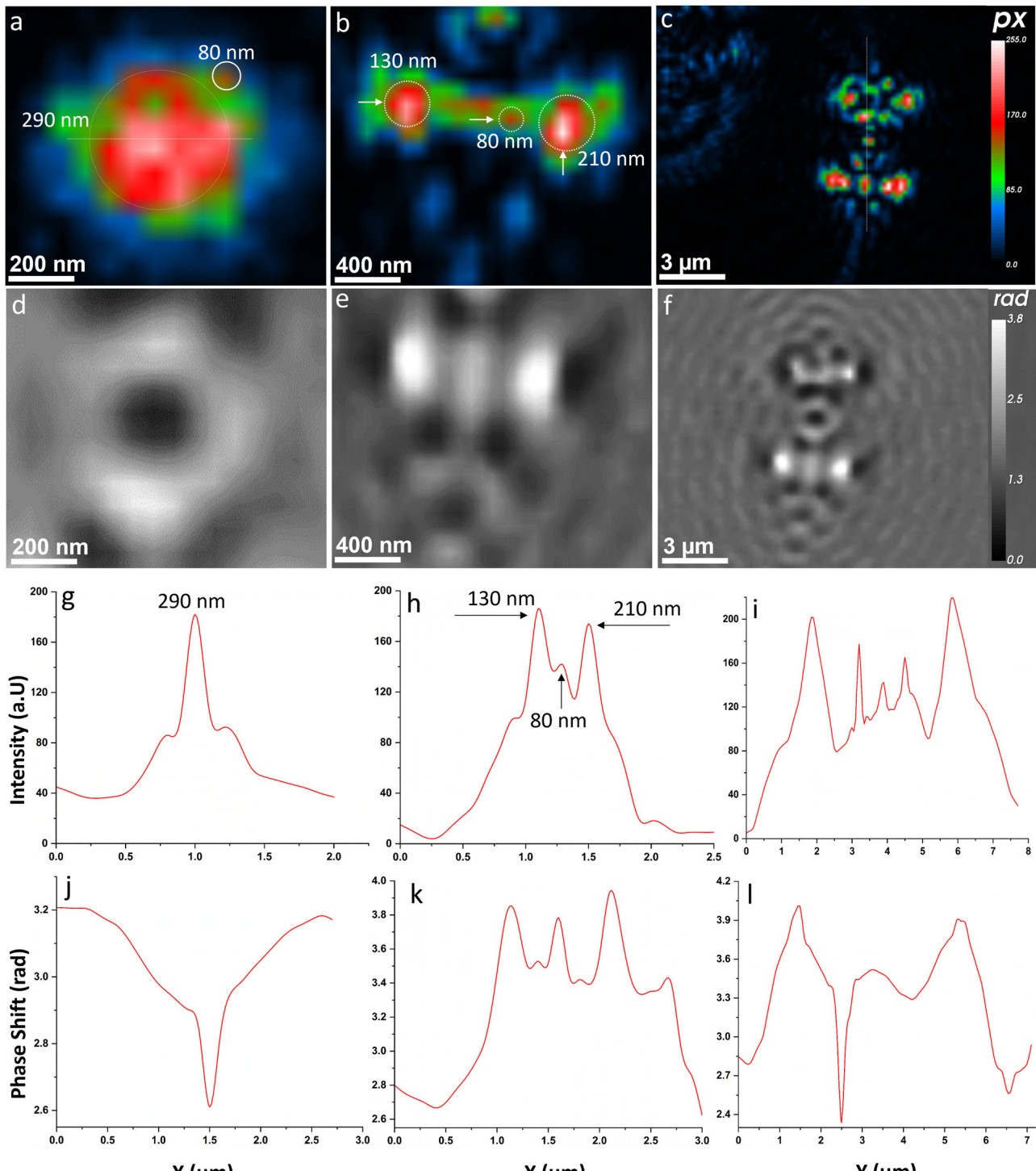

**Fig. 2 Intensity and phase reconstruction of airborne MS2. a–c** Is intensity reconstruction, and (**d–f**) presents phase reconstruction. (a) Intensity reconstruction of airborne MS2 viruses at Z = 627 μm, (b) Z = 687 μm, and (c) Z = 1570 μm. **d–f** Phase results of the same particles as (**a–c**). **g–i** Intensity crosscut profiles of particles in sections (**a–c**) and (**j–l**) phase crosscut profiles of particles in (**d–f**). The 4D dynamic trajectories of the particles are provided in Supplementary Movie 1 and Supplementary Movie 2. The background holograms recorded for zero air and particle concentrations tested by the SMPS and the OPS are shown in Supplementary Figure S4. The automated detection of airborne MS2 by Stingray software is shown in Figure S5.

transformation of airborne viruses, which potentially occurs in the case of SARS-CoV-2[33,41,45–47].

**4D trajectories of MS2 particles and droplets in air.** The dynamic 4D trajectories (3D positions and 1D time) of MS2 viruses were obtained by nano-DIHM of both dry MS2 aerosols and MS2 viral droplets in moving air, as shown in Fig. 3 and Supplementary Movies 1 and 2. Movie 1 presents the dynamic trajectories of MS2 viral droplets (sneezing model), and Movie 2 displays the trajectories of dry MS2 aerosols in moving air.

As depicted in Supplementary Movies 1 and 2, the dark red MS2 particles were in the focus of the reconstruction plane, while

**Table 1 3D size distribution of MS2, SARS-CoV-2 and MS2 mixed samples with TiO$_2$ in a single hologram using Octopus software.**

| Bacteriophage MS2 (dry aerosol) Statistics (µm) | Mean | Std | Median | 99th | 1st |
|---|---|---|---|---|---|
| Width | 0.25 | 0.97 | 0.18 | 1.40 | 0.04 |
| Height | 0.25 | 0.97 | 0.18 | 1.40 | 0.03 |
| Length | 0.50 | 0.40 | 0.38 | 1.74 | 0.06 |
| Bacteriophage MS2 + TiO$_2$ (dry aerosol) | | | | | |
| Width | 0.46 | 0.43 | 0.34 | 2.00 | 0.02 |
| Height | 0.45 | 0.43 | 0.33 | 1.99 | 0.02 |
| Length | 0.83 | 0.67 | 0.66 | 2.90 | 0.06 |
| Bacteriophage MS2 (droplet) | | | | | |
| Width | 1.20 | 3.70 | 0.77 | 6.29 | 0.07 |
| Height | 1.09 | 2.15 | 0.73 | 5.82 | 0.06 |
| Length | 1.40 | 1.11 | 1.12 | 4.76 | 0.10 |
| SARS-CoV-2 (Sneezed/viral droplets) Q1 = 10 m.sec$^{-1}$ | | | | | |
| Width | 0.63 | 0.68 | 0.45 | 3.15 | 0.03 |
| Height | 0.59 | 0.69 | 0.41 | 3.15 | 0.02 |
| Length | 0.72 | 0.72 | 0.60 | 3.89 | 0.01 |
| Q2 = 20 m.sec$^{-1}$ | | | | | |
| Width | 0.75 | 0.90 | 0.51 | 4.23 | 0.03 |
| Height | 0.72 | 0.85 | 0.49 | 3.79 | 0.03 |
| Length | 0.72 | 0.74 | 0.61 | 3.54 | 0.01 |
| Q3 = 30 m.sec$^{-1}$ | | | | | |
| Width | 0.82 | 1.32 | 0.56 | 4.18 | 0.04 |
| Height | 0.82 | 1.43 | 0.55 | 4.29 | 0.04 |
| Length | 1.99 | 1.73 | 1.58 | 5.51 | 0.04 |

The temporal resolution of 31.25 ms. The samples were modeled as a dry aerosol stream, droplets, or coughed-sneezed phase. Std means standard deviation.

some of the particles (green/blue) were slightly out of focus for a given reconstruction position. This occurred due to the finite depth of field of objective[39]. Multiple reconstructions of the same hologram in many planes are required to bring those out-of-focus particles into their dynamic trajectories[36,48]. As seen in Movie 1 and Fig. 3a–d, the MS2 particles were focused. However, many particles were progressively out of focus, indicating that the particles were also moving perpendicular to the reconstruction plane. This indicates that MS2 particles were in random motion, which also occurs in the case of aerosol particles[44]. The dynamic trajectories of MS2 particles in moving viral droplets (Fig. 3a) indicated morphology evolution over time, and the MS2 particles aggregated or coagulated after some time. This may be related to the MS2 particle interactions with water or the self-assembly of MS2 particles.

**In situ real-time SARS-CoV-2 detection, classification, and physicochemical characteristics**. Next-generation nano-DIHM[39] is innovated through the development of libraries and classifiers for several airborne nonvirus airborne particles as controls. Thus, we implemented artificial intelligence to distinguish and improve the accuracy of the physicochemical characterization of viral droplets and aerosols from other aerosols in the matrix[39,49]. Figs. 4 and 5 present for the first time in situ real-time observations of airborne SARS-CoV-2 detection using nano-DIHM. SARS-CoV-2 samples were obtained from the Department of Medicine at McGill University and were heat-inactivated (methods, Fig. S1b–d). PCR analysis was performed at the Department of Medicine, McGill University, to confirm the SARS-CoV-2 particles in the samples. BLASTN, using the beta coronavirus genomic database, is the result of the sequenced genome of the SARS-CoV-2 sample, as shown in Fig. S1a. The

RIM-1 viral stocks were whole-genome sequenced, and the GenBank accession number is MW599736. The physicochemical properties of these samples are presented in the next section.

**SARS-CoV-2 detection in dynamic mode**. Previously, in the literature, the transmission of viral droplets and aerosols produced during coughing and sneezing upon microbial infections has been studied[50,51], and the lifetimes of virus-laden droplets have been assessed. We performed experiments using these literature studies[51,52] to mimic three different sneezing/coughing types: air velocities ~10 m.sec$^{-1}$, 20 m.sec$^{-1}$, and 30 m.sec$^{-1}$. SARS-CoV-2-laden droplets were generated by using a c-flow atomizer. The droplets directly passed through the flow tube cuvette into the nano-DIHM sample volume, and holograms were recorded. The size data of the SARS-CoV-2 droplets are given in Table 1. Fig. 4a–l displays the intensity and phase results of SARS-CoV-2 droplets in dynamic mode, while Fig. 5a–f depicts the SARS-CoV-2 viral particle results in a stationary manner.

Examples of the intensity and phase reconstruction for airborne droplets of SARS-CoV-2 are shown in Fig. 4a–l. Fig. 4a shows that a raw hologram was recorded for the airborne droplets of SARS-CoV-2 particles. In contrast, the background hologram (Fig. 4b) was recorded without SARS-CoV-2 particles, and only purified dry air was used with three HEPA filters to serve as a control. Before processing the final reconstruction, the background hologram was subtracted from the raw hologram. This process is performed to remove possible contamination due to the pinhole and sample holders, such as a flow tube cuvette or microscope slide[39]. The contrast hologram shown in Fig. 4c is a product of subtracting the background hologram from the raw hologram. The intensity and phase reconstruction performed on the rectangular cropped area in Fig. 4c, as shown in Fig. 4d, is of interest. Fig. 4e, f is an example of the high-resolution intensity images in Fig. 4d, which was achieved by performing in-focus reconstruction, thus enhancing image quality and reducing noise[39]. Fig. 4f is a zoomed-in area of Fig. 4e, revealing a SARS-CoV-2 shape that is well matched to that of prior studies using high-resolution microscopy[46,53,54]. Phase-reconstructed images of the identical hologram exemplified in Fig. 4g–i yield a similar form of SARS-CoV-2 particles as the intensity images (Fig. 4d–f). The intensity and phase reconstruction showed that the SARS-CoV-2 cluster size was 400, 600 nm, 1.2 µm and 3.4 µm.

As depicted in Fig. 3a, b, under similar experimental conditions, the coagulation of several viral particles was observed, or, more likely, the vesicles were covered with moist water vapor (since the experiment was performed in droplet form). Previous studies suggested that SARS-CoV-2 viral-laden particles vary from submicron to several-micrometer particles[11,55] due to water or organic/inorganic compound uptake in the environment[33]. Nano-DIHM provided a similar size trend and trajectory analysis (Movie S3), confirming the water uptake and morphological change.

The 3D size distribution (Tables 1), 3D orientation and individual dimensions of SARS-CoV-2 viral particles are shown in Fig. S7. The 3D size data of SARS-CoV-2 depicted variations from ~300 nm to several micrometers (Table 1). The 3D size data of the SARS-CoV-2-laden vesicles/droplets/aerosols obtained by nano-DIHM were in agreement with the virus sizes found due to coughing, sneezing, and breathing in previous studies (Table S4). The mean/median values of the SARS-CoV-2 viral droplet width increased with higher velocities (Table 1). This is due to SARS-CoV-2 droplet aggregation or coagulation.

The 4D dynamic trajectories of the SARS-CoV-2 droplet motion are shown in Supplementary Movie 3. The trajectory analysis successfully illustrated the morphological evolution of

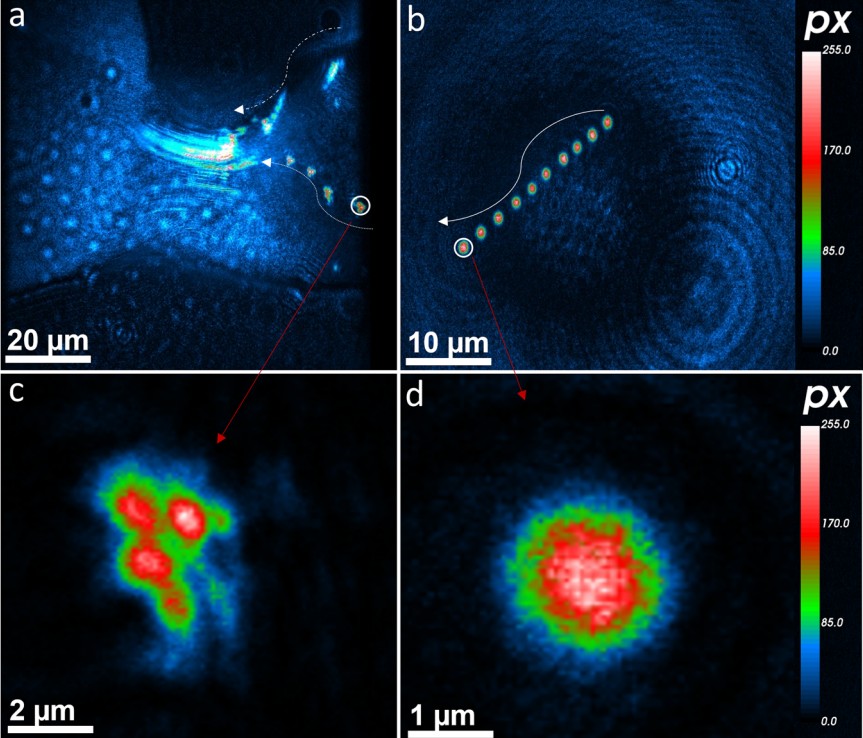

**Fig. 3 Trajectory analysis of MS2 particles in droplet form and dry aerosols. a** MS2 viral-laden droplet particle trajectories. The trajectories were obtained by the sum of 13 holograms with 31.25 ms temporal resolution. **b** Trajectories of MS2 particles in dry aerosols were obtained with the sum of 10 holograms with 31.25 ms temporal resolution. **c**, **d** Zoomed-in particles from their trajectories. The white arrows indicate the direction of particle motion. The blue interference pattern suggests that particles are not focused.

SARS-CoV-2 particles over time. The SARS-CoV-2-laden particles moved randomly. The morphological changes of the SARS-CoV-2 vesicle/droplet/aerosols were well aligned with the existence of multiple variants of SARS-CoV-2[47,53,56–58]. Observing real-time surface properties and their morphological changes is vital to quickly responding to not only SARS-CoV-2 but also any future pandemics caused by unknown viruses or other microbial entities.

**SARS-CoV-2 detection in stationary mode.** The confirmation of SARS-CoV-2 particle size, phase, shape, and morphology in a stationary manner is displayed in Fig. 5. The observation of SARS-CoV-2 particles made by nano-DIHM (Fig. 5a–f) was in agreement with the images made by the 10X magnification AMG Evos XL core microscope (Fig. S1b–d). The intensity and phase results showed that the SARS-CoV-2 particle size varied from 1 micron to several micrometers (Figs. 5 and S8). Additional intensity and phase profiles along the crosscut of the particle are shown in Fig. S8. The positive phase shift varied from 3.1 to 4.1 radians across the particles (Fig. S8d), suggesting water uptake. In contrast, the negative phase shifting from 3.2 to 2.6 rad may indicate the different sites of SARS-CoV-2 particles (Fig. S8c). In this study, nano-DIHM clearly shows advantages over other optical microscopy methods[54] because of its simple configuration and in situ real-time measurement capabilities in terms of the size, shape, phase, and morphology of viral entities, which is not possible using optical microscopy.

**Building a library and classifiers for fully automated SARS-CoV-2 detection (Yes/No).** Stingray software was trained to achieve the real-time in situ automatic detection and physicochemical characterization of SARS-COV-2 using nano-DIHM. The Stingray software working and training procedure is given in

methods. We trained the Stingray software for multiple sample matrices (Table S2) in dynamic and stationary modes. In this study, Stingray software training was performed using over 10000 holograms and 100 K iterations to achieve an accuracy of approximately 99% for identifying SARS-CoV-2 or any targeted particles in a mixed sample in air and water. The workflow of training process of Stingray given in Table 3. For instance, Table 2 presents the automated detection of SARS-CoV-2 in the mixed MS2 sample with "YES" and "NO" outcomes. Table S3 provides an example of the physicochemical characteristics (size, shape, and surface morphology) of viruses and metal oxide particles.

The accuracy of automated detection and classification by Stingray software may be decreased based on sample matrix complexity. Nevertheless, this can be addressed by building an extensive library of multiple sample matrices. Nano-DIHM may also give false positives. Further improvement of classifiers and surface data will likely reduce this disadvantage.

**Physicochemical characteristics of MS2 with TiO₂ and organic coating.** We also performed a series of experiments to explore whether nano-DIHM enables deciphering a coating suite of naturally observed organic and inorganic/metallic particles on MS2 viruses. We examined highly viscous droplets, such as honey ($C_6H_{12}O_6$), olive oil ($C_{88}H_{164}O_{10}$), and alpha-pinene ($C_{10}H_{16}$), mixed with MS2 or with TiO₂ and PSL (mostly $C_8H_8)_n$). The results of TiO₂ and oil-coated MS2 viruses are shown in Figs. 6, S9, and S10. The coating impact of alpha-pinene and honey on MS2 viruses is depicted in Fig. S11.

The intensity and phase results of TiO₂ particles are shown in Fig. 6a, b and c, d, representing the TiO₂-coated MS2 viruses. Fig. 6e, f displays the electron microscopy images of TiO₂-coated MS2 viruses. The intensity crosscuts across particle 1 (Fig. 6a) and particles 1 and 2 (Fig. 6c) are shown in Fig. 6m–p. The intensity

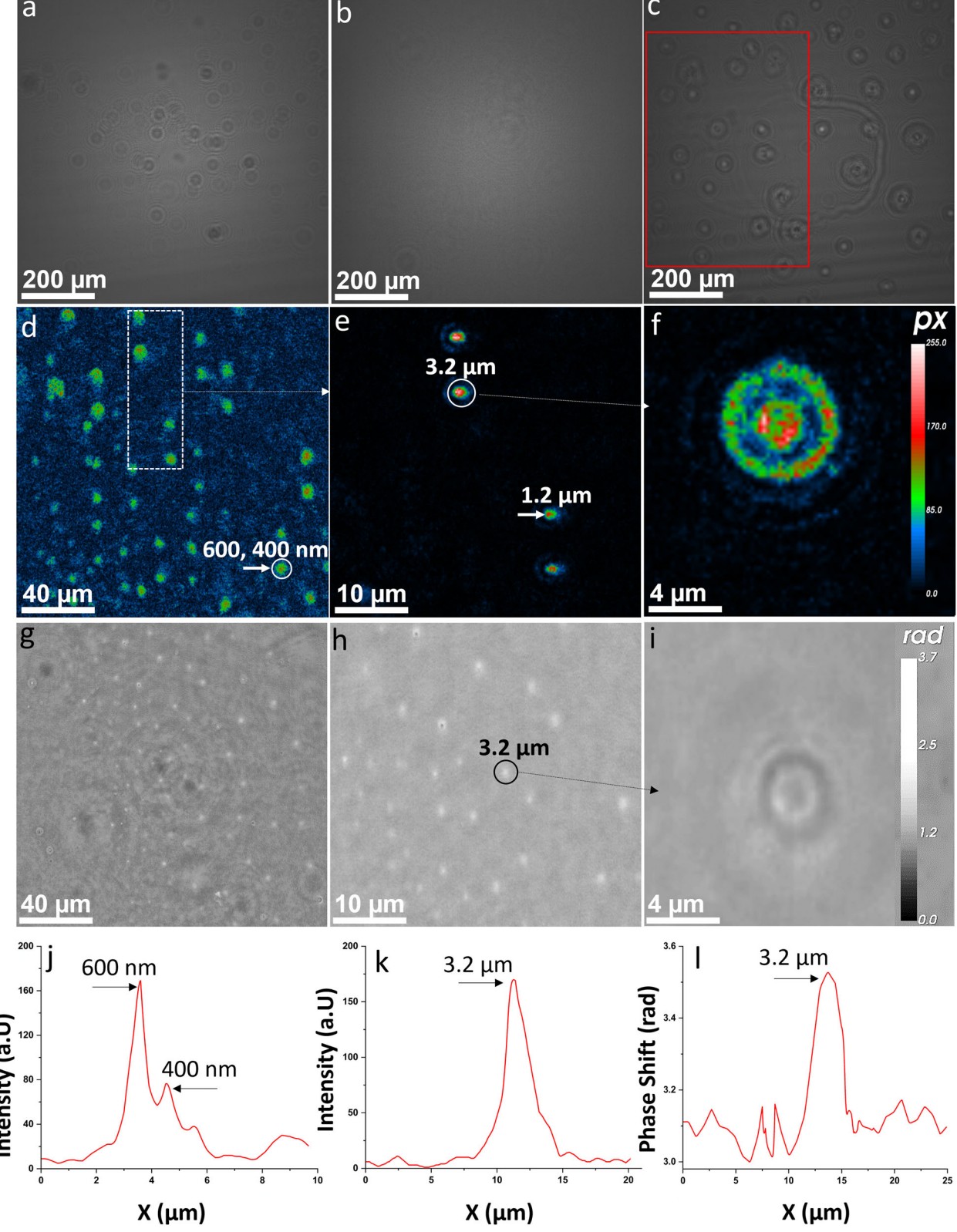

**Fig. 4 Inactivated SARS-CoV-2 droplet detection by nano-DIHM. a** Raw hologram recorded for SARS-CoV-2 viral droplet particles. **b** Background hologram recorded without particles. **c** Contrast hologram obtained after subtracting the background hologram from the raw hologram. **d** Zoomed-in area of (**c**) at Z = 2109 μm. **e** Zoomed-in area of (**d**) revealing the precise recovery of SARS-CoV-2 viral droplets and their shape. **f** A more focused zoomed-in image of (**e**) clearly demonstrates the SARS-CoV-2 droplet structure. **g–i** Phase reconstruction of identical SARS-CoV-2 particles. **j**, **k** Intensity profile of PSL particles across the particle crosscut in (**e**). **l** Phase profile of the SARS-CoV-2 particle crosscut in (**h**).

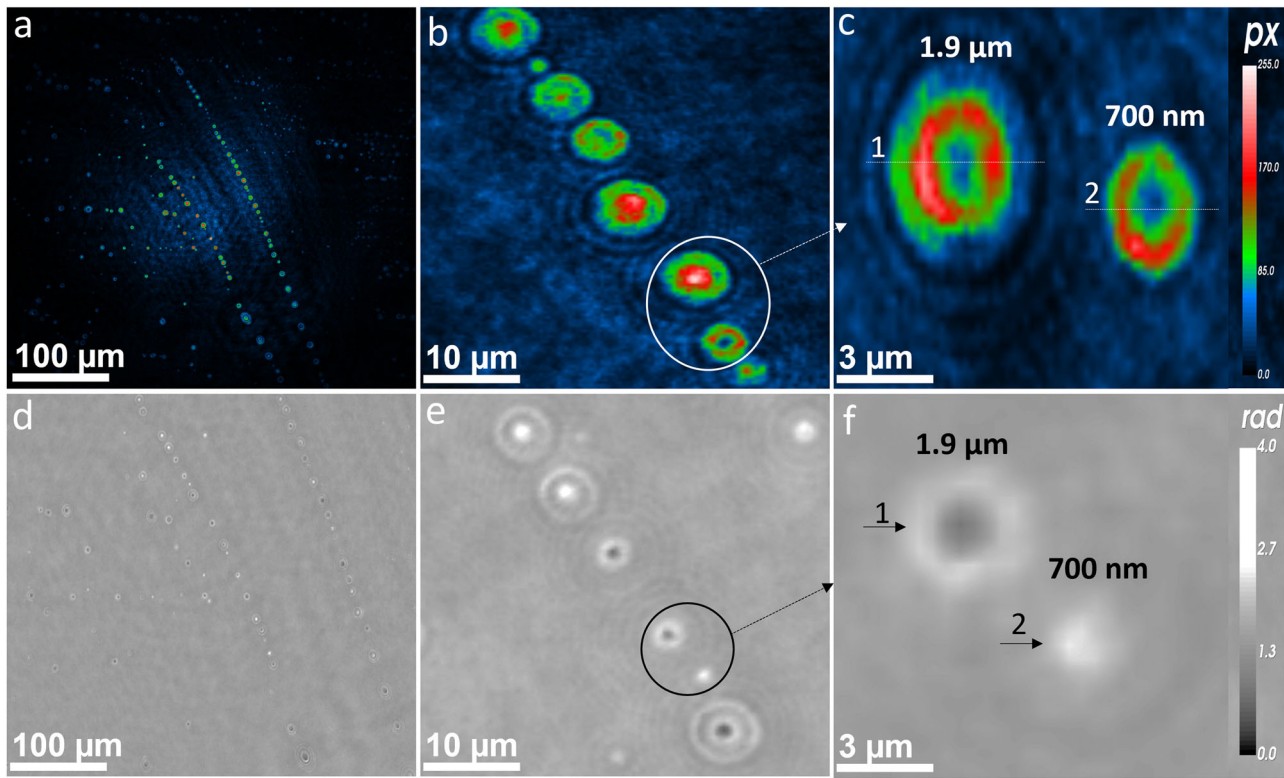

**Fig. 5 SARS-CoV-2 detection in stationary mode by nano-DIHM. a–c** Intensity images of SARS-CoV-2 particles and (**d–f**) phase reconstructions. **c** The in-focus SARS-CoV-2 viral-laden particles circled in (**b**). Legends 1 and 2 in (**c**) show the size of SARS-CoV-2 viral-laden particles/droplets. **f** displayed the optical phase of SARS-CoV-2 viral-laden particles circled in (**e**). The black arrow and legends 1 and 2 in (**f**) indicate the size of SARS-CoV-2 viral-laden particles/droplets. The intensity and phase crosscut of particles 1 and 2 in (**c**) and (**f**) are shown in Supplementary Figure S8.

profiles of $TiO_2$-coated MS2 viruses demonstrated the attachment of MS2 viruses on $TiO_2$ or vice versa and enhanced the size from nanosized to several micrometers (Fig. 6m–p, Fig. S9). The high-resolution electron microscopy images confirmed the attachment of $TiO_2$ to MS2 viruses, likely due to the high surface tension of the MS2 and $TiO_2$ particles. More intensity and phase reconstruction images of $TiO_2$-coated MS2 viruses are shown in Fig. S9. The intensity and phase reconstruction images of pure olive oil are shown in Fig. 6g, h, and those for olive oil-coated MS2 virus particles are shown in Fig. 6i–j. High-resolution electron microscopy images of olive oil-coated MS2 virus are displayed in Fig. 6k, l. Interestingly, the olive oil-coated MS2 virus tended to adhere to contact-type morphologies more than $TiO_2$-coated MS2 viruses. This may be due to the adhesive properties of olive oil. More intensity and phase results of olive oil-coated MS2 viruses are shown in Fig. S10.

Fig. S11a–d illustrates the intensity and phase images of alpha-pinene (secondary organic aerosols)-coated MS2 viruses. More-over, the honey-coated MS2 particle intensity and phase results are shown in Fig. S11e, f. The multimodal intensity and phase shift crosscut distribution observed in the alpha-pinene-coated MS2 particles indicated the surface modification or heterogeneity of the samples. However, honey-coated MS2 particles maintained their original structures with elongated/agglomerated morphologies.

Nano-DIHM successfully distinguished the coating impact on MS2 particles (Figs. 6, S9–11) with distinct pure MS2 particles (Fig. 1). It is clear evidence that alpha-pinene had a more vital interaction with the MS2 surface and transformed the MS2 particle structure into a layered structure. In contrast, $TiO_2$ was attached to the MS2 viruses, enhancing the MS2 size and altering its surfaces and morphologies. The olive oil and honey coatings altered MS2 morphologies due to their strong viscosity and

adhesive surfaces. We further provided an example of iron oxide and PSL coating on MS2 particles, as presented in Fig. S12. Overall, nano-DIHM offers promising results for providing real-time in situ physicochemical virus characterization.

**Surface properties of SARS-CoV-2, MS2, and metal oxide particles**. To investigate the surface morphology of SARS-CoV-2 particles, we also imaged MS2, 200-nm polystyrene latex spheres (PSLs), olive oil, and metal oxide nanoparticles ($TiO_2$ and iron oxide) by using nano-DIHM. As shown in Table S3, the edge gradient and surface roughness of SARS-CoV-2 particles were quite distinct from those of PSL, olive oil, MS2 bacteriophage, and metal oxide particles. Note that reference holograms were obtained between experiments, including those with HEPA fil-tering. However, we cannot overrule possible contaminants such as impurities in Milli-Q water. However, as we developed clas-sifiers for each item, the contaminants were detected and subtracted. The apparent difference in the edge gradient of SARS-CoV-2 and MS2 is due to their different surface properties. SARS-CoV-2 was heat-inactivated, while MS2 was an active virus. To demonstrate the structured exposure of viruses obtained by nano-DIHM, we exposed the MS2 samples to UV-B light (280-315 nm) for 30 minutes before the experiment. The MS2 particle shape with and without UV-B was similar, but the edge gradient of UV-B-exposed MS2 was observed to be almost half that of the unexposed MS2. Fig. S5 shows the automated detection of MS2 particles using Stingray with UV-B and without UV-B. The real-time observation of surface roughness, size, phase, and time-dependent changes in the morphology of the SARS-CoV-2 in an ambient environment could be a significant breakthrough in understanding the physical process for not only SARS-CoV-2 but also future unknown viruses.

**Table 2 Yes/No detection of SARS-CoV-2 from mixed samples using Stingray software.**

| No | Yes | Yes | Yes | No | Yes | No |
|----|-----|-----|-----|----|-----|----|

| No | No | No | Yes | No | Yes | Yes |
|----|-----|-----|-----|----|-----|----|

A mixed sample of SARS-CoV-2 and MS2 particles was analyzed. "YES" indicates SARS-CoV-2, and "NO" indicates MS2 particles. This automated classification is in progress, and we have built full automation for SARS-CoV-2 and will develop it for future viruses, metals, plastics, and bacteria. Table S3 discusses the surface properties of multiple sample matrices.

## Conclusion

The currently developed nano-DIHM[39] can detect, classify, and determine the physicochemical properties of SARS-CoV-2 in air and water in the blink of an eye. The Nano-DIHM is a portable unit (Fig. S13) and can act as a virus sensor, like a breathalyzer or an aerosol analyzer. Nano-DIHM can operate in static/dynamic mode at the site or laboratory to produce results in less than a minute with an accuracy of +90%. In contrast, conventional testing methods for COVID−19 are expensive and time-consuming; none are in situ or real-time methods. (Table S1). A future promising feature of nano-DIHM is that it can allow simultaneous measurements of several types of even more diverse particles that could signal both active and past infections from multiple viruses.

An increase in microbial pandemic occurrences is expected due to climate change[59]. The capability of nano-DIHM to determine the in situ and real-time physicochemical transformation of viruses and other pollutants and contaminants, such as nano- and microplastics and nanometals, would provide an edge over existing technologies. The real-time tracking of SARS-CoV-2 or any future viruses allows policymakers to react swiftly with more knowledge in future epidemic management responses.

In brief, nano-DIHM can be used in a broad range of research and technology, from time-dependent physical and chemical transformation of viruses and other microbiological entities, biogeochemistry, noninvasive imaging, biophysics, life cycle analysis of environmental pollutants, sustainable technology and pharmaceutical-medicinal applications to space and climate change science.

## Methods

**Digital in-line holographic microscopy**. Digital in-line holographic microscopy (DIHM) works as a two-stage process: 1) recording the holograms and 2) numerically reconstructing the holograms to yield object(s) information. In the current setup, the holograms are recorded using the 4Deep Desktop Holographic Microscope (Halifax, Nova Scotia)[34]. Numerical reconstruction is performed using the improved Octopus software, version 2.2.2[39,60] and the improved Stingray software package, version 2.2.2[61].

The detailed theory of DIHM and the reconstruction process are given in our previous paper[39]. In brief, a schematic of the next-generation nano-DIHM setup is displayed in Fig. 7. A pinhole (laser (L)) emits a wave at $\lambda = 405$ nm. The resulting wave illuminated objects and produced a highly magnified diffraction pattern (hologram) on a screen[36,39]. A complementary metal-oxide semiconductor (CMOS) sensor records holograms and stores them on a computer for subsequent numerical reconstruction[34,60].

Fig. 7b shows that light emitted from the pinhole propagates toward the screen and is scattered by the particles/objects in its way, resulting in a hologram. The wave amplitude of the hologram on the screen, $A(r, t)$, is given by Eq. 1.

$$A(r, t) = A_{ref}(r, t) + A_{scat}(r, t) \quad (1)$$

where $A_{ref}(r, t)$ and $A_{scat}(r, t)$ are the reference and scattered amplitudes, respectively.

The resultant intensity of the hologram recorded on the screen is:

$$I(r, t) = A(r, t)A^*(r, t)$$

$$I(r, t) = A_{ref}(r, t)A^*_{ref}(r, t) + [A_{ref}(r, t)A^*_{scat}(r, t) + A_{scat}(r, t)A^*_{ref}(r, t)] + A_{scat}(r, t)A^*_{scat}(r, t) \quad (2)$$

In Eq. 2, the first term represents the beam's intensity in the absence of an object or scatterer, and the last term represents the intensity of the scattered wave. The second term in the square brackets indicates the interferences between the reference and the scattered waves, referred to as holograms. The amplitude of the scattered hologram is:

$$A_{scat}(r) = \frac{iA_{ref}}{r\lambda} \iint I(r) \frac{\exp\left(ik\frac{rr'}{r}\right)}{|r - r'|} ds \quad (3)$$

During the numerical reconstruction of holograms, only three parameters are required to yield the object information: 1) the distance between the source (pinhole) and the screen, 2) the wavelength of light ($\lambda = 405$ nm), and 3) the camera pixel size (5.5 μm)[39]. Our experiment shows the quality of the background holograms in Fig. S4a. The background hologram was recorded with purified air. The airborne particles exiting the gas flow tube cuvette are passed into the SMPS and the OPS through the nano-DIHM sample volume (Fig. 7d). The particle counts measured with the SMPS and the OPS for purified air are fewer than 2 particles/cm³ (Fig. S4).

The nanosized resolution was obtained using specific experimental and numerical reconstruction approaches[39]. First, the hologram was recorded at the tip of the pinhole, keeping a minimum distance between the sample and the source. This procedure enables higher magnification, and hence a higher resolution can be achieved. Further, we modified the Octopus and Stingray software by implementing the additional convolution-deconvolution route to achieve a higher resolution. The details of those approaches can be found in our previous paper[39]. The Octopus and Stingray software can be operated online and offline mode. The Octopus and Stingray software can simultaneously record and analyze the holograms with a temporal resolution of 31.25 ms in real-time. The Stingray software analysis can also be performed remotely in the laboratory or at home by accessing real-time recorded holograms files which can be stored on clouds/one drive or any other data depository source. The operation mode (online or offline) of Stingray software does not impact the performance of the software or Nano-DIHM.

**Building library: automation and classification process**. The automation and classification of SARS-CoV-2 viral-laden droplets were performed using Stingray software (Table 3, flow chart in the manuscript). The Stingray software is based on a patented algorithm[37,38] that was trained to achieve the real-time in situ automatic detection, classification, and physicochemical characterization of SARS-COV-2 in situ in real by using nano-DIHM. The stingray software workflow follows three main categories: 1) Identify and find objects from recorded or real-time holograms, 2) classify the objects into taxon, and 3) start training the classifiers. The basic algorithm of stingray software follows the Kirchhoff-Fresnel reconstruction approach[37,38], including robust deep neural network classifiers that extract in-focus objects and

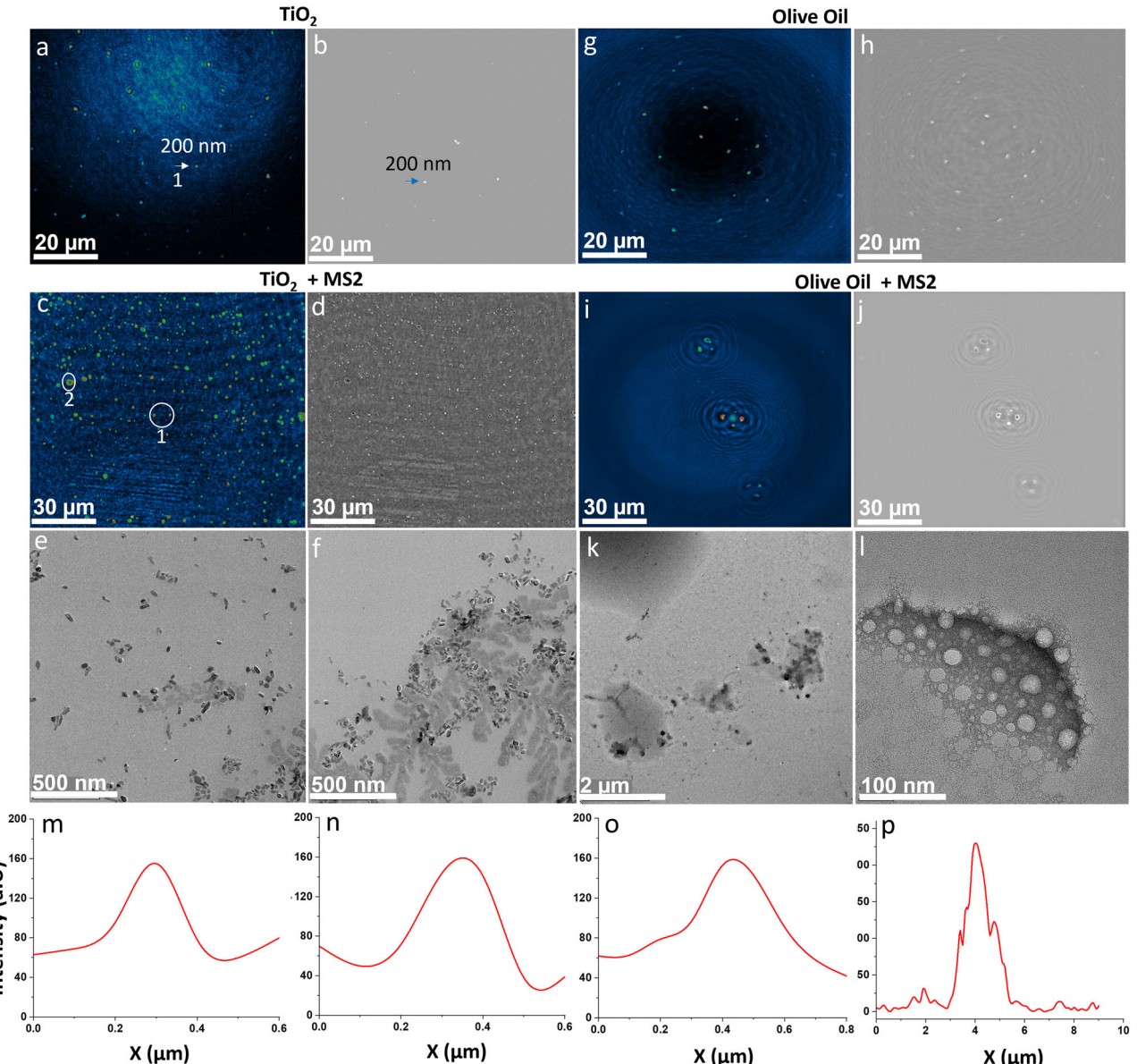

**Fig. 6 MS2 viruses coated with TiO₂ and olive oil. a, b** Intensity and phase reconstruction of TiO₂ particles. The white and green arrow indicates the TiO₂ particles and the intensity crosscut of the TiO2 particle is represented by the white arrow shown in (**m**). **c, d** Intensity and phase results of MS2 viruses coated with TiO₂ particles obtained by nano-DIHM. Legends 1 and 2 displayed the particles within white circles, and their intensity crosscuts were shown in (**n–o**) and (**p**), respectively. **e, f** High-resolution electron microscopy images of TiO₂-coated MS2 viruses. **g, h** Intensity and phase images of olive oil. **i, j** Intensity and phase results of MS2 viruses coated with olive oil obtained by nano-DIHM. **k, l** High-resolution electron microscopy images of oil-coated MS2 viruses. **m** Intensity response of particle 1 in (**a**). **n–p** Intensity response of particles 1 and 2 in (**c**), respectively.

classify them within the image volume[61]. The intensity threshold and edge gradient value will be used as input parameters to detect the virus or other objects.

The 10 K hologram and 100k iteration were used to train the Stingray software, and 99% accuracy was achieved. The following steps were performed to identify and classify SARS-CoV-2, MS2 and other materials (Table S2): 1) the input of holograms and recording parameters, such as the camera pixel size, laser wavelength and source-to-camera distance; 2) the optimization of intensity threshold values, which is responsible for finding the particles within the threshold domain. A good threshold value can be achieved by reconstructing the hologram manually using Octopus software, and 3) choosing and selecting SARS-CoV-2-laden particles and classifying them into groups. This classification is based on the shape/morphology, intensity threshold and edge gradient. The flexibility of ± 5% of their threshold allowed them to be classified or identified as particles/viruses/materials. Once the threshold parameter is optimized, the Stingray software can detect and classify the objects from millions of holograms. The automated outcome results contain object information, including the sizes, roughness, edge gradient, surface area, and shape of the particles. This process can be performed for both static and dynamic samples.

Table 1 shows the automated classification and detection of several materials and their associated physical properties, such as their size, shape and surface properties. Table 2 shows the ability of Stingray software to classify and identify SARS-CoV-2 from the mixed samples, and the outcome of "YES" indicates SARS-CoV-2 and "NO" indicates MS2. To validate the accuracy, we used seven different types of classifiers to compare the results. They are included: MS2 (dry aerosols), MS2 (moist droplets), TiO₂ (dry aerosols), 200 nm PSL (dry aerosols), SARS-CoV-2 + SARS-CoV-2 RNA + TiO₂ (water), SARS-CoV-2 + MS2 (air and water). The major challenge of the accuracy of Stingray software may decrease depending on the complexity of the sample matrix. A more extensive library of multiple sample matrices is required to overcome this issue. The extended/extensive library also allowed us to identify or target unknown species. Since several known or unknown species exist in the natural environment/atmosphere, Nano-DIHM cannot extract information on unknown species without information on targeting species. We have shown that Nano-DIHM successfully detected and classified the oil spills in water samples[49]. The next generation nano-DIHM may detect the unknown particles if they are viruses or not. Since even unknown viruses have physicochemical characteristics, we may have the rapid training of the software and confirmation with more conventional PCR techniques in future.

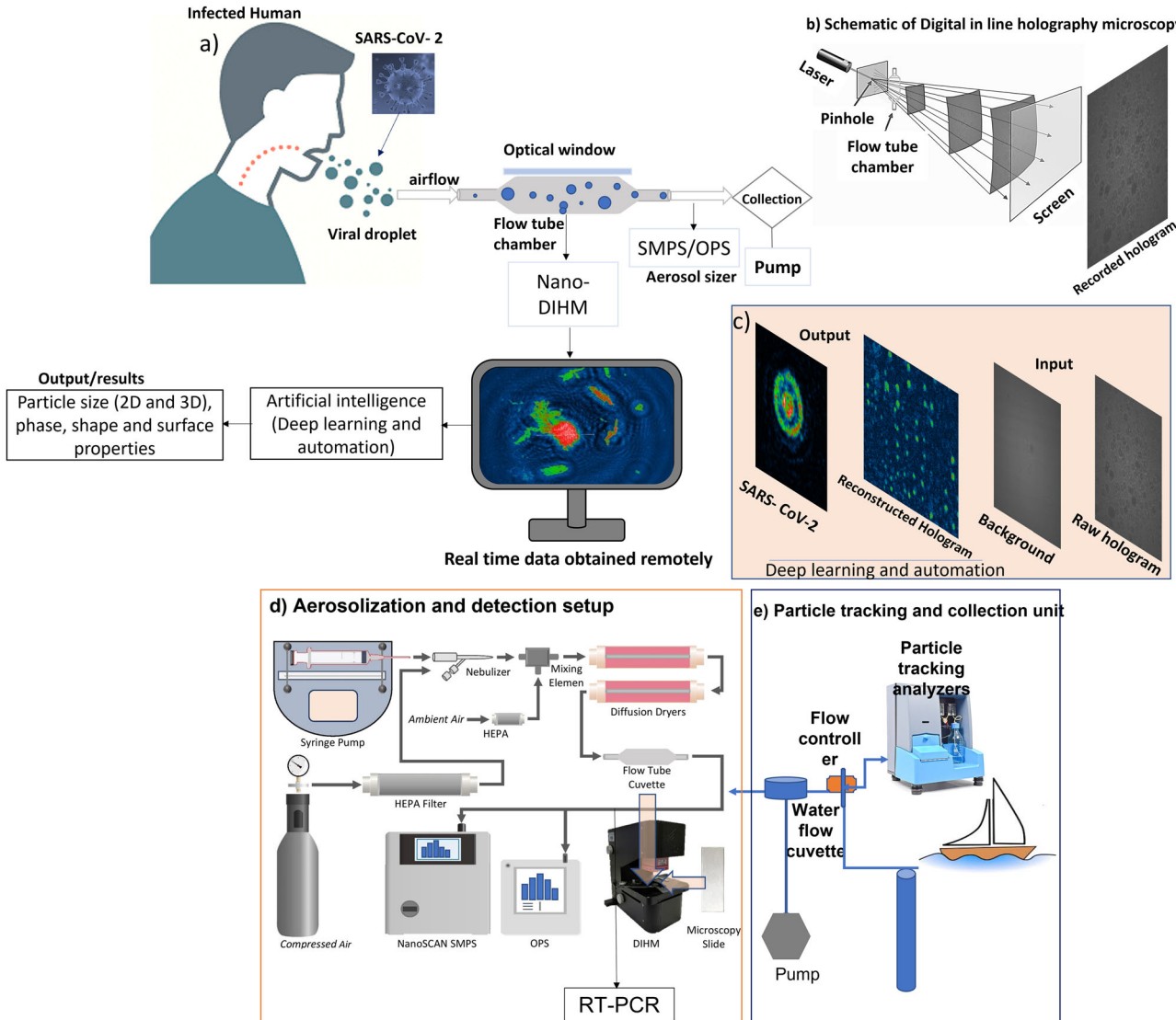

**Fig. 7 Schematic of nano-DIHM setup. a** SARS-CoV-2 transmission by an infected human via airborne transmission. The airborne viral droplets were passed through the flow tube cuvette to the nano-DIHM sample volume and Scanning Mobility Particle Sizer (SMPS). The nano-DIHM was used to record the airborne viral droplets, and Octopus/Stingray software (artificial intelligence) was used to detect and characterize viral particles. **b** Working principle of holography microscopy, where laser/pinhole emits the light and holograms are recorded on the screen. **c** An example of Deep learning for SARS-CoV-2 analysis, where raw and background holograms are input images and Stingray software determines the SARS-CoV-2 physicochemical properties. **d** The experimental setup of airborne particle characterization at the laboratory. This experimental setup is versatile and can characterize synthetic objects' aerodynamic behavior by aerosolizing them using a syringe pump and atomizer. And **e**) Particle tracking analysis and sample collection method for field data.

**SARS-CoV-2 sample information**. We obtained heat-inactivated SARS-CoV-2 samples from the Medicine Department at McGill University. The heat inactivation process was performed at 92 °C for 20 minutes by shaking the samples. To confirm the SARS-CoV-2 particles in the samples, the Facility of Medicine Department at McGill University performed an RT‒PCR test, and the genome sequencing results of the SARS-CoV-2 sample are given in Fig. S1. The GenBank ID for the sequence is MN908947.3. Furthermore, SARS-CoV-2 sample images were obtained under the 10X magnification of an AMG Evos XL core microscope (Fig. S1b–d) before supplying the samples for nano-DIHM measurements.

The bacteriophage MS2 samples ($1.0 \times 10^9$ pfu/ml) were purchased from ZeptoMetrix and stored below −20 °C until they were used for the experiment. MS2 sample preparation, such as dilution and mixing with metal oxides, was performed under a clean biosafety fume hood. Nano-DIHM requires no prior sample preparation for imaging virus particles and measuring combined organics or metal oxide particles. The original bacteriophage MS2 samples were diluted by a volume of 100x before performing the nano-DIHM measurements.

**Experimental setup**. A schematic of the integrated experimental setup of next-generation nano-DIHM is shown in Fig. 7. The experimental design (Fig. 7) for the measurement of airborne viral particles consisted of the following components: 1)

DIHM instrument, 2) gas flow cuvette (ES Quartz Glass, volume of 700 µL, path length 2 mm), 3) microscope slide (Quartz Glass), 4) aerosol generator unit, 5) aerosol sizers and 6) sample collection unit for further analyses. Our previous papers provide a detailed description of the aerosol generation unit, aerosol particle sizers[31], and nano-DIHM[39]. The airborne/waterborne viruses (bacteriophage MS2 and SARS-CoV-2) passed through the quartz flow tube cuvette, and holograms were recorded by nano-DIHM (Fig. 7a–e).

Several sample matrices were tasted in both dynamic and stationary manners. The detailed sample information and recording parameters are given in Table S2. The holograms were recorded for the moving airflow (containing viral aerosols) stream passed through the gas flow cuvette installed in nano-DIHM with a final flow rate of 1.7 L/minute. The outflow (1.7 L/min) from the cuvette was connected to the SMPS and the OPS. The coupling of the SMPS and the OPS with the DIHM allowed the determination of the aerosol size distribution of the particles imaged by DIHM in situ in real-time. In addition, the mixed samples with MS2 and $TiO_2$ were used to examine the heterogeneity and physiochemical transformation of viral particles in the air. We also combined SARS-CoV-2, SARS-CoV-2 RNA and iron oxide particles and analyzed them directly by nano-DIHM. Nano-DIHM determined the size, shape, and morphology of the bacteriophage MS2 compared with the morphology visualized using S/TEM.

**Table 3 Workflow diagram for automation and classification of SARS-CoV-2 using Stingray software.**

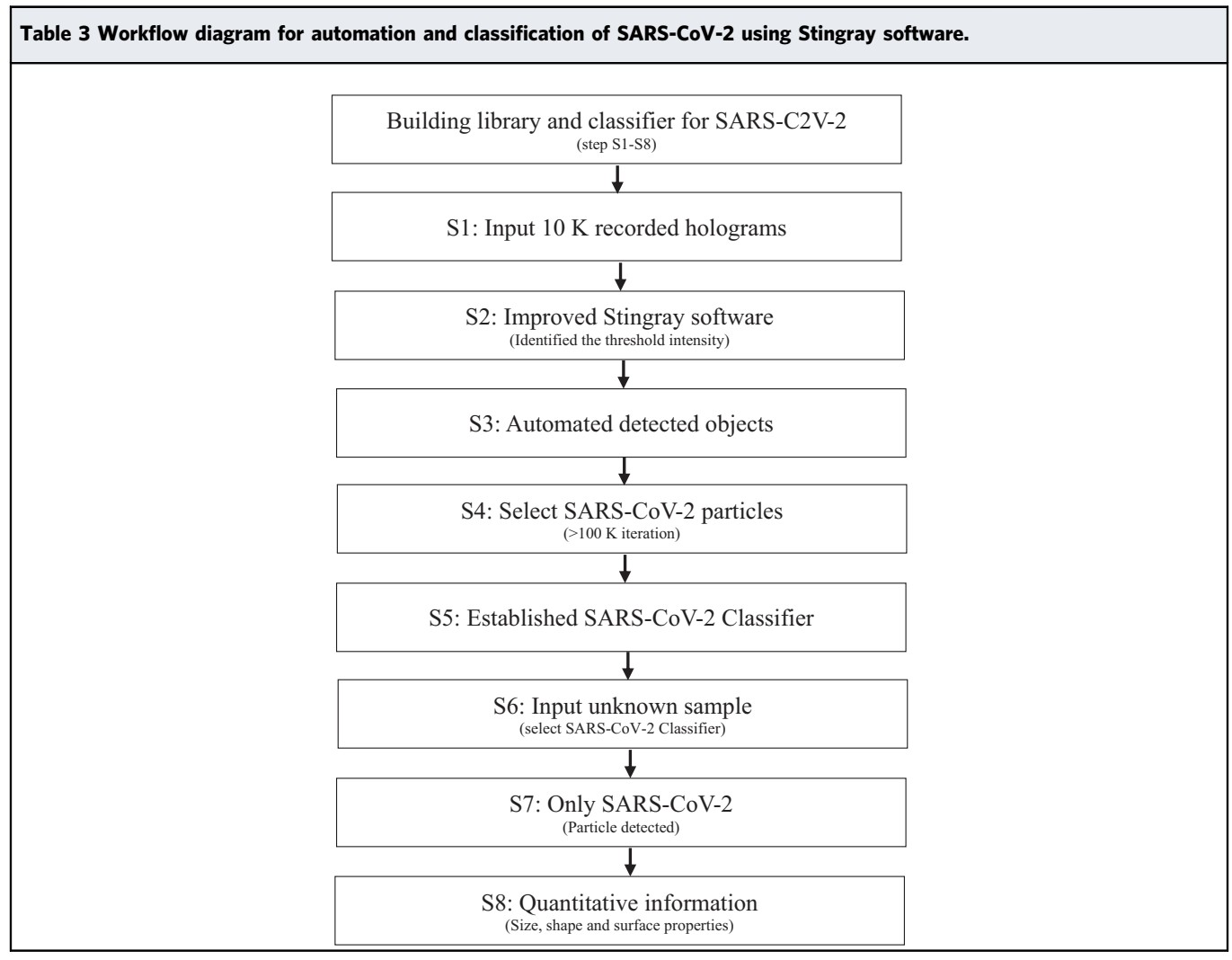

We also performed the experiments in stationary mode. To do that, 15 µL sample drops were placed on a microscope slide using a micropipette. Once the sample was placed on the slide, a cover slide was used. Furthermore, sample-loaded microscope slides were placed in a nano-DIHM sample holder, and images were recorded.

**Aerosol sizers**. In this study, a scanning mobility particle sizer (NanoScan™ SMPS model 3910, TSI Inc.) and an Optical Particle Sizer (OPS, model 3330, TSI Inc.) was used to measure the real-time size distributions of airborne particles[31,39]. The SMPS measured the particle sizes in the range of 10 nm to 400 nm, and the OPS determined the particle size in a range of 0.3 µm – 10 µm. The sample flow rate for the SMPS was 0.75 L/min, while the OPS required a sampling flow rate of 1 L/min. A more detailed description of the SMPS and the OPS system is provided in our previous articles[31,62].

**High-resolution electron microscopy**. First, the TEM grid was negatively charged using a 15 MA plasma glowing discharge, and later, 10 µL drops of pure MS2 bacteriophage solution were applied to the grid. After 5 minutes, the excess MS2 sample was removed using filter paper. Afterward, uranyl acetate (aq. 2%, w/v) was applied for negative staining. One minute later, the excess uranyl acetate was removed with filter paper. Electron microscopy images were taken by using a transmission electron microscope. Thermo Scientific Talos F200X G2 S/TEM with ChemiSTEM technology, including an X-FEG high brightness Schottky field emission Source, Ceta 16 M 4k x 4k CMOS camera, super-x windowless energy dispersive spectrometer, and gatan enfinium ER Model 977 electron energy loss spectroscopy (EELS), were used with a couple of high visibility low-background beryllium double-tilt optimized for EDS.

S/TEM (Tecnai G$^2$F20 S/TEM microscope) was used to analyze the 100X diluted MS2 samples in Milli-Q water. For S/TEM analysis, no staining was performed. A 10 µL drop of 100X MS2 samples was applied to the TEM grid and allowed to remain for 1 minute. Afterward, TEM grids were placed onto the sample holder, and TEM images were acquired. A detailed description of S/TEM is given in our previous paper[31,39].

**Litesizer particle analyzer**. A Litesizer 500 (Anton Paar, Canada) particle sizer analyzer (PSA) was used to characterize the active MS2 virus sizes in 100x diluted MS2 samples in aqueous mode. The Litesizer 500 measures the particle size from nm to micrometers. The Litesizer measures the particle sizes via dynamic light scattering at three different measurement angles: side, back, or forward scattering, allowing optimal parameter settings.

**Particle trajectory analysis**. The following procedure was used to achieve high-resolution trajectories: (a) a series of holograms were recorded at 32 fps for both moving air and flowing water in a quartz cuvette, (b) the experimental/optical impurity of the background was eliminated by subtraction of consecutive holograms, and (c) the resultant holograms were reconstructed at a particular reconstruction position (plane) and summed to obtain the dynamic trajectories[39]. The subtraction of holograms was necessary to ensure that the dynamic range was not exceeded and only the MS2 virus information was preserved[39,48]. All the holograms were reconstructed at the same reconstruction position (Z = 5409 µm) for moving viral droplets (Movie S1) and Z = 1790 µm for moving air (Movie S2). Furthermore, reconstructed results were processed to create the Giff movies.

## Data availability

The data that support the findings of this study are available from the corresponding author upon reasonable request.

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

## Acknowledgements

We are very grateful to several colleagues at McGill Faculty of Medicine and Health Sciences, particularly the virologists, including Ms. Fiona McIntosh, Professor Marcel Bher, and Professor Jorg Fritz, who provided us with inactivated viruses and guided us with the procedures. We thank Professor Vali from the McGill Facility of electron microscopy for the S/TEM analysis. The Tomlinson Award and McGill Sustainability supported this work to PAA, Canadian Foundation for Innovation (CFI), Natural Sciences and Engineering Research Council of Canada (NSERC), National Research Council (NRC), NSERC CREATE PURE, and PRIMA Quebec. We also thank Dr. Amit Kumar Pandit, National Institute of Aerospace, Hampton, Virginia, USA, for his critical review of the manuscript and Mr. Ryan Hall for proofreading our manuscript.

## Author contributions

D.P. and P.A.A. designed and performed the experiments. D.P. performed the data analysis and drafted the manuscript. P.A.A. supervised D.P. and revised the manuscript. C.L. and M.A. provided critical constructive suggestions. P.A.A. wrote the scientific proposal that was the basis of this study and funded this project.

## Competing interests

The authors declare no competing interests.
