## [Peer Review File · Communications Engineering]

Real-time 4D tracking of airborne virus-laden droplets and aerosolsReviewers' comments:

Reviewer #2 (Remarks to the Author):

The authors report about an interdisciplinary study on the characterization of airborne particles with a commercial digital in-line holographic microscope. Artificial particles, bacteriophages, inactivated SARS-CovV-2, RNA fragments, as well as mixtures including metallic and organic materials, are analyzed for size and morphology under static and dynamic conditions and are classified utilizing a commercial software. The results are compared to complementary data from electron microscopy.

In general, the study is motivated and includes adequate references. The experimental investigations appear to be accurately performed and results are novel. The proposed analysis concept represents a contribution to the enhancement of the current state-of-the-art in airborne particle analysis. The authors address an important topic: The label-free detection and identification of virus-like particles which may be of interest for the area of particle analysis and digital in-line holographic microscopy. In summary, the content of the manuscript appears to be suitable for the journal Communications Engineering.

However, the authors should consider revisions:

1. *Title*: The study is performed label-free with digital in-line holographic microscopy which appears to be an essential topic. This should be reflected also by the title.
2. *Abstract*: As the study seems to be performed with a commercial digital in-line holographic microscopic and commercial software this should be mentioned / emphasized in the abstract.
3. *Introduction* (lines 82-91): From the descriptions in the introduction the general measurement principle of the utilized digital in-line holographic microscopy (DIHM), which seems to be the central topic of the manuscript becomes not clear for an interested interdisciplinary reader. Moreover, no information is provided how nanoscopic resolution can be achieved, and why the combination with artificial intelligence is beneficial for the measurement procedure, and not any supporting reference concerning these topics is provided. Substantial additional clarifying information should be provided, and, for clarity, the authors may consider suppressing “nano” in “nano-DIHM”.
4. *Methods*:
 - a. Neither in the descriptions in the main text nor from the explanations in the methods section distinct information is provided that / if a commercial in-line digital holographic microscope is applied in the study and if it is a lens-based or a lens-less instrument. The authors should clarify this topic in both parts of the manuscript. If it is a commercial instrument the manufacturer and type of the instrument should be specified with more details. Addition of further clarifying information is required.
 - b. Classification algorithms: The authors write that for particle classification the commercial “stingray” software is applied and trained. However, no information about the underlying classification algorithms and not any supporting reference seems to be provided. Without information about the applied

classification algorithm the experimental results are not convincing. The authors should add further

information about the software, the utilized classification algorithms and should add a number of suitable supporting references to clarify these topics.

5. *Results and discussion*: The descriptions of the experimental results are partly difficult to understand for an interdisciplinary reader. The authors provide very detailed descriptions of the main results. However, the main results / essential information / conclusions / outcomes become not fully clear. To clarify the presentation in the results section the authors should consider:

a. Adding a figure (e.g., like Fig. S1 in the supplementary information) with a simplified sketch of the entire experimental arrangement and the workflow of the overall sample preparation and the measurement process at the beginning of the results section, e.g., as the first figure (Fig. 1) and adequate explanation instead of only referencing to the supplementary material.

b. Revision of all figures and figure captions for clarity:

i. E.g., by indication of the different imaging modalities (intensity, phase electron microscopy, etc., in or near the images within the figure) as the images look partly very similar, and by emphasizing / mentioning the main results / outcomes in the first sentence of the explaining figure caption, like for example: "Size and morphology of MS2 particles in DHIM images agree with electron microscopy image data (for Fig. 1), or "Trajectory analysis of MS2 analysis in droplet form indicates that (for Fig. 3) ",..., same for Figs. 2, 4, 5,6.

ii. Reduction of tables 1 and 2 to representative / illustrating results and shift all other data to an appendix or to the supplementary data.

iii. Providing a summary of the main outcomes of the results / the main conclusions at each subsection of results section.

6. *Discussion*: The results-section descriptions include many details, but the main findings as well as their impact/ consequences become not fully clear to the reader (see also comment 5). For clarity, the authors should separate the section discussion from the results section and should discuss their results in a more extended manner also by considering the significance, the possible impact, and the possible limitations of the proposed particle analysis concept.

Minor points:

- Tables 1 and 2: Information about the measurement uncertainty should be provided, e.g., by calculation of the standard deviation.

- Figures 2g-f: The unit of y-axis "intensity" seems to be missing ("a. U." or "gray levels"?). Clarification is required.

- Figures 2m-p: The meaning of "Number density" becomes not clear (particles / cm³). Clarification is required.

- Plots in Figures 2g-l, 4j-l, 6m-p: The labeling is very tiny and should be enlarged for improved visibility.

- Flow chart in line 378: The provided information seems to be a list and thus could be also appear as a listing in the main text.

Reviewer #3 (Remarks to the Author):

Four-dimensional in situ real-time physicochemical tracking of virus-laden droplets and aerosols in air

D. Pal, M. Amyot, C. Liang, P.A. Ariya

This manuscript reports on an apparatus that purports to perform real-time virus tracking using nano-digital in-line holographic microscopy. The authors claim to be able to characterize the time-dependent state of viruses and other particulates with respect to a variety of physicochemical properties.

Line 161: “While nano-DIHM determined and classified...” this statement ends with the assertion that it did not perform as well as high-resolution S/TEM; however, it’s just not possible. DIHM cannot perform as well because of the different wavelength scales involved. This reads as if the authors expect that they could achieve the same resolution with a little more work. Be clear that it is not possible.

Table 1: What are these images? Are these reconstructions or holograms? Some look like reconstructed images, while others have the look and feel of a hologram. Clarify.

Line 212: When discussing the size of the MS2 particles be sure to make clear that there are aggregates of the viral particles.

Line 247: Figure 5 is cited out of order. Reference the figures in the order in which they appear.

Line 261: What is the depth of field of the objective?

Line 348: Quantify real-time.

Regarding the classifier, what is being used? It appears to be a binary (YES/NO) classification routine. What data is in the reference database? How many different types is the classifier comparing the know samples? What is the characteristic that most often triggers a successful classification? How long is this whole process? How large is the database? What is large enough to achieve accurate classification?

Line 473: The authors state, “...nano-DIHM will be used...” Will it? This is a bold statement. Use the more conservative “may be used” or “can be used.”

Line 486: Mentions the setup, but there is no accompanying figure. Include a detailed apparatus schematic in the manuscript at the appropriate point.

Line 511: SMPS and OPS are, I believe, used as acronyms before they are defined.

Line 603: "A more extensive..." How much more extensive? How many more samples? The experiments are somewhat limited in this regard, and the authors have not been clear in the manuscript.

There is an inordinate amount of text that refers to the supplemental material, including references to other figures, etc. If the material is so important that it is referenced in the manuscript then it should be in the manuscript. There should not be this much reliance on this extra material. Supplements are great locations for things like video or code.

The paper is all right. It needs substantial work to get to a form that is suitable for publication. At this point, the authors should revisit the manuscript and clarify the points that need to be made more clear for the reader. Also, revise the manuscript partitioning so there is less reliance on the supplemental section.

Prof. Parisa Angeline Ariya

Department of Chemistry and Department of Atmospheric and Oceanic Sciences

McGill University, 801 Sherbrooke St. West Montreal, PQ, Canada

E-mail: parisa.ariya@mcgill.ca

RE: Revision of the manuscript entitled "Four-dimensional in situ real-time physicochemical tracking of virus-laden droplets and aerosols in the air "

March 14, 2023

To: Reviewers,

Communications Engineering: Nature

Dear Reviewers/colleagues,

Please find an attached revised manuscript entitled "*Four-dimensional in situ real-time physicochemical tracking of virus-laden droplets and aerosols in the air*" by Devendra Pal, Marc Amyot, Chen Liang and Parisa A. Ariya for consideration for publication in your journal.

We sincerely thank you for your time, suggestions, and consideration. We very much appreciate critical comments and suggestions. We carefully considered reviewers' comments and suggestions and have made significant changes to the manuscript accordingly. In the enclosed documents, you will find a point-by-point response letter to each comment or suggestion made by the reviewers, and the corresponding modifications made to the manuscript. We trust that we have now addressed all their comments and have made a substantial improvement to the manuscript.

Thanks very much in advance for your time and consideration. If you require any further information, please feel free to contact me.

Cordially,

Parisa A. Ariya

James McGill Chair of Chemistry and Atmospheric and Oceanic Sciences

Reply

We cordially thank the reviewers and the editor for their time, contributions, and comments. We truly appreciate them and trust that they are satisfied with our modifications that address all their concerns. For your convenience, the reviewer's comments are italicized, and our responses are in regular font.

Reviewers' comments:

Reviewer # 2 (Remarks to the author):

"The authors report about an interdisciplinary study on the characterization of airborne particles with a commercial digital in-line holographic microscope. Artificial particles, bacteriophages, inactivated SARS-CovV-2, RNA fragments, as well as mixtures including metallic and organic materials, are analyzed for size and morphology under static and dynamic conditions and are classified utilizing a commercial software. The results are compared to complementary data from electron microscopy. In general, the study is motivated and includes adequate references. The experimental investigations appear to be accurately performed and results are novel. The proposed analysis concept represents a contribution to the enhancement of the current state-of-the-art in airborne particle analysis. The authors address an important topic: The label-free detection and identification of virus-like particles which may be of interest for the area of particle analysis and digital in-line holographic microscopy. In summary, the content of the manuscript appears to be suitable for the journal Communications Engineering."

Response: Thank you very much for your appreciation and positive remarks for the study's novelty. We would also like to thank you for your critical comments and suggestions. We know that they are very constructive and appreciate it.

"1. Title: The study is performed label-free with digital in-line holographic microscopy which appears to be an essential topic. This should be reflected also by the title."

Response: Thank you very much for your remark. Indeed, the study was performed with upgraded holographic microscopy, a label-free technology.

- We opted not to include label-free or holographic microscopy in title because we have upgraded the holographic microscope in terms of both optical operation and numerical reconstruction algorithm; hence we call it Nano-DIHM, as we described elsewhere ^{1,2}.
- However, as per your suggestion, we have mentioned it clearly in the abstract. As it can be read in lines 24-26:

"Here, we provide evidence for 4-dimensional physicochemical tracking of virus-laden droplets and aerosols in the air using automated next-generation label-free nano-digital in-line holographic microscopy (Nano-DIHM)."

- For accuracy, we opted not to include label-free in the "title" as we have improved the regular holographic microscope on several fronts; both optical operation and we incorporated the AI (deep

learning)-numerical reconstruction algorithm; hence we call it Nano-DIHM, as we described in detail elsewhere^{1,2}.

"2. Abstract: As the study seems to be performed with a commercial digital in-line holographic microscopic and commercial software this should be mentioned / emphasized in the abstract."

Response: Thank you very much for the suggestion.

- Indeed, the holograms were recoded using a Desktop Holographic Microscope (4Deep, Halifax, Nova Scotia), and numerical reconstructions were performed using modified Octopus/stingray software (4Deep, Halifax, Nova Scotia).
- We have improved optical operation and the addition of the capability to be operated in both dynamic and stationary modes that allow in-situ and real-time 4D detection of airborne viruses.
- However, the method section provides the commercial details and working principles of DIHM. It is given in lines 495-544.
- Furthermore, we have made additional modification and performed the suite of changes in the original Desktop Holographic Microscope (4Deep, Halifax, Nova Scotia) and its operation process. We also modified the software described in our previous paper¹. Thereby, we used the term Nano-DIHM in our abstract.
- Please kindly note that no commercially available DIHM equipment is capable of performing 4D tracking of airborne viruses and providing physicochemical characteristics in situ and real-time for particles less than 200 nm, except for current works.

"3. Introduction (lines 82-91): From the descriptions in the introduction the general measurement principle of the utilized digital in-line holographic microscopy (DIHM), which seems to be the central topic of the manuscript becomes not clear for an interested interdisciplinary reader. Moreover, no information is provided how nanoscopic resolution can be achieved, and why the combination with artificial intelligence is beneficial for the measurement procedure, and not any supporting reference concerning these topics is provided. Substantial additional clarifying information should be provided, and, for clarity, the authors may consider suppressing "nano" in "nano-DIHM."

Response: Thank you very much for your suggestion. We have now included brief information on DIHM in the introduction, given in lines (82-90). It can read as:

“This study used Nano-Digital in-line Holographic Microscopy (Nano-DIHM) to investigate the viruses in air and water in situ in real time. The Nano-DIHM comprises a desktop holographic microscope (4Deep, Halifax, Nova Scotia)³ and a gas flow tube that allows airborne particles to travel through the imaging volume of the DIHM, enabling real-time observation of single or ensembles of viral particles or other objects. Nano-DIHM is a lensless technology that directly records interference patterns called holograms of the incident and scattered light using a light-sensitive matrix/digital camera³⁻⁵. The object information was recovered from the recorded holograms by performing numerical reconstruction using Octopus/Stingray^{6,7} software based on a patented algorithm^{8,9}.”

- In short, we used a combination of experimental and numerical reconstruction approaches to reach the nanosized resolution. First, the hologram was recorded at the tip of the pinhole, keeping a minimum distance between the sample and the source. This procedure enables higher magnification, and hence a higher resolution can be achieved. Further, we modified the Octopus and Stingray software by implementing the additional convolution-deconvolution route to achieve a higher resolution. The details of those approaches can be found in our previous paper ¹. This statement has been included in method section.
- The details of the working principle of DIHM and the experimental procedure are given in the method section from line 495-544.

"4. Methods: a. Neither in the descriptions in the main text nor from the explanations in the methods section distinct information is provided that / if a commercial in-line digital holographic microscope is applied in the study and if it is a lens-based or a lens-less instrument. The authors should clarify this topic in both parts of the manuscript. If it is a commercial instrument the manufacturer and type of the instrument should be specified with more details. Addition of further clarifying information is required."

Response: Thank you very much for your comment.

- Indeed, the holograms were recoded using a commercial Desktop Holographic Microscope (4Deep, Halifax, Nova Scotia), and numerical reconstructions were performed using modified Octopus/stingray software (4Deep, Halifax, Nova Scotia).
- Please note, we have modified and performed the suite of changes in the original Desktop Holographic Microscope (4Deep, Halifax, Nova Scotia) and software described in our previous paper ¹.
- For further clarification, the schematic figure was provided in supporting information Figure S1 moved to the revised manuscript as Figure 7.
- As suggested earlier, we have also included brief information on Nano-DIHM in the introduction. However, the Method Section now describes Nano-DIHM and experimental procedures.

As per your convenience, Nano-DIHM information is given in the Method section in lines 495-544 and it can be read as:

Digital in-line holographic microscopy

“Digital in-line holographic microscopy (DIHM) works as a two-stage process: 1) recording the holograms and 2) numerically reconstructing the holograms to yield object(s) information. In the current setup, the holograms are recorded using the 4Deep Desktop Holographic Microscope (Halifax, Nova Scotia)³. Numerical reconstruction is performed using the improved Octopus software, version 2.2.2 ^{1,6} and the Stingray software package, version 2.2.2 ⁷.

The detailed theory of DIHM and the reconstruction process are given in our previous paper¹. In brief, a schematic of the next-generation nano-DIHM setup is displayed in Figure 7. A pinhole (laser (L)) emits a wave at $\lambda = 405$ nm. The resulting wave illuminated objects and produced a highly magnified diffraction pattern (hologram) on a screen^{1,4}. A complementary metal-oxide semiconductor (CMOS) sensor records holograms and stores them on a computer for subsequent numerical reconstruction^{3,6}.

Figure 7 (b) shows that light emitted from the pinhole propagates toward the screen and is scattered by the particles/objects in its way, resulting in a hologram. The wave amplitude of the hologram on the screen, $A(r, t)$, is given by Equation 1.

$$A(r, t) = A_{ref}(r, t) + A_{scat}(r, t) \quad (1)$$

where $A_{ref}(r, t)$ and $A_{scat}(r, t)$ are the reference and scattered amplitudes, respectively.

The resultant intensity of the hologram recorded on the screen is:

$$I(r, t) = A(r, t)A^*(r, t)$$

$$I(r, t) = A_{ref}(r, t)A_{ref}^*(r, t) + [A_{ref}(r, t)A_{scat}^*(r, t) + A_{scat}(r, t)A_{ref}^*(r, t)] + A_{scat}(r, t)A_{scat}^*(r, t) \quad (2)$$

In Equation 2, the first term represents the beam's intensity in the absence of an object or scatterer, and the last term represents the intensity of the scattered wave. The second term in the square brackets indicates the interferences between the reference and the scattered waves, referred to as holograms. The amplitude of the scattered hologram is:

$$A_{scat}(r) = \frac{iA_{ref}}{r\lambda} \iint I(r) \frac{\exp\left(ik \frac{rr'}{r}\right)}{|r - r'|} ds \quad (3)$$

During the numerical reconstruction of holograms, only three parameters are required to yield the object information: 1) the distance between the source (pinhole) and the screen, 2) the wavelength of light ($\lambda = 405$ nm), and 3) the camera pixel size ($5.5 \mu\text{m}$)¹. Our experiment shows the quality of the background holograms in Figure S4(a). The background hologram was recorded with purified air. The airborne particles exiting the gas flow tube cuvette are passed into the SMPS and the OPS through the nano-DIHM sample volume (Fig. 7(d)). The particle counts measured with the SMPS and the OPS for purified air are fewer than 2 particles/cm³ (Fig. S4).

The nanosized resolution was obtained using specific experimental and numerical reconstruction approaches¹. First, the hologram was recorded at the tip of the pinhole, keeping a minimum distance between the sample and the source. This procedure enables higher magnification, and hence a higher resolution can be achieved. Further, we modified the Octopus and Stingray software by implementing the

additional convolution-deconvolution route to achieve a higher resolution. The details of those approaches can be found in our previous paper ¹.”

"b. Classification algorithms: The authors write that for particle classification the commercial "stingray" software is applied and trained. However, no information about the underlying classification algorithms and not any supporting reference seems to be provided. Without information about the applied classification algorithm the experimental results are not convincing. The authors should add further information about the software, the utilized classification algorithms and should add a number of suitable supporting references to clarify these topics."

Response: Thank you for your comment. Indeed, the automated classification was performed using Stingray software (4deep, Halifax, Nova Scotia). The Octopus and Stingray software are based on a patented algorithm ^{8,9}. To improve the accuracy and resolution, we modified the Octopus and Stingray software by implementing the additional convolution-deconvolution routine to achieve a higher resolution. The details of those approaches can be found in our previous paper ¹.

As per your suggestions, we have provided the detailed working principle of Stingray in the revised manuscript in Line number 545-584. It can read as:

Building library: automation and classification process

“The automation and classification of SARS-CoV-2 viral-laden droplets were performed using Stingray software (flow chart in the manuscript). The Stingray software is based on a patented algorithm ^{8,9} that was trained to achieve the real-time in situ automatic detection, classification, and physicochemical characterization of SARS-COV-2 in situ in real by using nano-DIHM. The stingray software workflow follows three main categories: 1) Identify and find objects from recorded or real-time holograms, 2) classify the objects into taxon, and 3) start training the classifiers. The basic algorithm of stingray software follows the Kirchhoff-Fresnel reconstruction approach ^{8,9}, including robust deep neural network classifiers that extract in-focus objects and classify them within the image volume ⁷. The intensity threshold and edge gradient value will be used as input parameters to detect the virus or other objects.

The 10K holograms and 100k iterations were used to train the Stingray software, and 99% accuracy was achieved. The following steps were performed to identify and classify SARS-CoV-2, MS2 and other materials (Table S2): 1) the input of holograms and recording parameters, such as the camera pixel size, laser wavelength and source-to-camera distance; 2) the optimization of intensity threshold values, which is responsible for finding the particles within the threshold domain. A good threshold value can be achieved by reconstructing the hologram manually using Octopus software, and 3) Choosing and selecting SARS-CoV-2-laden particles and classifying them into groups. This classification is based on the shape/morphology, intensity threshold and edge gradient. The flexibility of $\pm 5\%$ of their threshold allowed them to be classified or identified as particles/viruses/materials. Once the threshold parameter is optimized, the Stingray software can detect and classify the objects from millions of holograms. The

automated outcome results contain object information, including the sizes, roughness, edge gradient, surface area, and shape of the particles. This process can be performed for both static and dynamic samples.

Table 1 shows the automated classification and detection of several materials and their associated physical properties, such as their size, shape and surface properties. Table 3 shows the ability of Stingray software to classify and identify SARS-CoV-2 from the mixed samples, and the outcome of “YES” indicates SARS-CoV-2 and “NO” indicates MS2. To validate the accuracy, we used seven different types of classifiers to compare the results. They are included: MS2 (dry aerosols), MS2 (moist droplets), TiO₂ (dry aerosols), 200 nm PSL (dry aerosols), SARS-CoV-2 + SARS-CoV-2 RNA + TiO₂ (water), SARS-CoV-2 + MS2 (air and water). The major challenge of the accuracy of Stingray software may decrease depending on the complexity of the sample matrix. A more extensive library of multiple sample matrices is required to overcome this issue. The extended/extensive library also allowed us to identify or target unknown species. Since several known or unknown species exist in the natural environment/atmosphere, Nano-DIHM cannot extract information on unknown species without information on targeting species. We have shown that Nano-DIHM successfully detected and classified the oil spills in water samples². The next generation nano-DIHM may detect the unknown particles if they are viruses or not. Since even unknown viruses have physicochemical characteristics, we may have the rapid training of the software and confirmation with more conventional PCR techniques in future.”

"5. Results and discussion: The descriptions of the experimental results are partly difficult to understand for an interdisciplinary reader. The authors provide very detailed descriptions of the main results. However, the main results / essential information / conclusions / outcomes become not fully clear. To clarify the presentation in the results section the authors should consider:

- a. Adding a figure (e.g., like Fig. S1 in the supplementary information) with a simplified sketch of the entire experimental arrangement and the workflow of the overall sample preparation and the measurement process at the beginning of the results section, e.g., as the first figure (Fig. 1) and adequate explanation instead of only referencing to the supplementary material."*

Response: Thank you very much for your suggestion.

- In the revised manuscript, we have simplified the text and explained the brief experiments for the reconstruction process in each result subheading. For further clarity, we have also moved the supplementary Figure S1 to the main manuscript (as Figure 7 in the method section) for further clarity.

"b. Revision of all figures and figure captions for clarity: i. E.g, by, indication of the different imaging modalities (intensity, phase electron microscopy, etc., in or near the images within the figure) as the images look partly very similar, and by emphasizing / mentioning the main results / outcomes in the first sentence of the explaining figure caption, like for example: "Size and morphology of MS2 particles in

DHIM images agree with electron microscopy image data (for Fig. 1), or "Trajectory analysis of MS2 analysis in droplet form indicates that (for Fig. 3)",..., same for Figs. 2, 4, 5, 6."

Response: Thank you very much for your suggestion. As for clarity for readers, we have now revised the Figures (2, 4 and 6) and their statements and captions. We have also simplified the text formulation in the revised manuscript.

"ii. Reduction of tables 1 and 2 to represent / illustrate results and shift all other data to an appendix or to the supplementary data."

Response: Thank you very much for your suggestions. Tables 1 and 2 are an integral part of the study. Table 1 provided an example of the automated detection of several objects and associated their physicochemical properties, including size, morphology, edge gradient and surface roughness. While table 2 provides the size statistics of different materials used in this study.

"iii. Providing a summary of the main outcomes of the results / the main conclusions at each subsection of results section."

Response: Thank you very much for your suggestion. In the revised manuscript, we have provided the closing statement/summary of the results in each section.

"6. Discussion: The results-section descriptions include many details, but the main findings as well as their impact/ consequences become not fully clear to the reader (see also comment 5). For clarity, the authors should separate the section discussion from the results section and should discuss their results in a more extended manner also by considering the significance, the possible impact, and the possible limitations of the proposed particle analysis concept."

Response: Thank you very much. Please kindly note that we followed the journal format, and it requires to have results and discussion together with the inclusion of subheadings. However, as per your suggestions, we have clarified the links between the paragraphs to further fluidity of the text.

Minor points:

"Tables 1 and 2: Information about the measurement uncertainty should be provided, e.g., by calculation of the standard deviation."

Response: Thank you. The standard deviation was included in Table 2 as SD. In the revised manuscript, we clarified in table caption and presented standards deviation as (Std) in the table 2.

"Figures 2g-f: The unit of y-axis "intensity" seems to be missing ("a. U." or "gray levels"?). Clarification is required."

Response: Thank you very much. In the revised manuscript, we have included it. Yes, it is a.U, and we have revised the axis level and included the intensity unit.

"Figures 2m-p: The meaning of "Number density" becomes not clear (particles / cm³?). Clarification is required."

Response: Thank you very much for your question. The number density in Figure 2(m-p) refers to the particle number concentration in unit air volume. This terminology is well-established in aerosol science 10-13.

"Plots in Figures 2g-l, 4j-l, 6m-p: The labeling is very tiny and should be enlarged for improved visibility."

Response: Thank you for your suggestions. We have enlarged the figures' labels and their axis font (Figures, 2, 4 and 6) in the revised manuscript.

"Flow chart in line 378: The provided information seems to be a list and thus could be also appear as a listing in the main text."

Response: Thank you very much. As per your suggestion, we further improved the text formulation in the revised manuscript. Detailed classification information was provided in the method section in line from Line number 545-584.

Reviewer #3 (Remarks to the author):

"This manuscript reports on an apparatus that purports to perform real-time virus tracking using nano-digital in-line holographic microscopy. The authors claim to be able to characterize the time-dependent state of viruses and other particulates with respect to a variety of physicochemical properties. There is an inordinate amount of text that refers to the supplemental material, including references to other figures, etc. If the material is so important that it is referenced in the manuscript then it should be in the manuscript. There should not be this much reliance on this extra material. Supplements are great locations for things like video or code.

The paper is all right. It needs substantial work to get to a form that is suitable for publication. At this point, the authors should revisit the manuscript and clarify the points the need to be made more clear for the reader. Also, revise the manuscript partitioning so there is less reliance on the supplemental Section."

Response: Thank you very much for your appreciation and positive remarks for the study. We would also like to thank you for your critical comments and suggestions. We know that they are very constructive and appreciate it.

- We herein addressed all your comments, rearranging the manuscript and supporting information.
- We moved Figure S1 from supplementary materials to the main revised manuscript as a Figure 7.
- Please kindly note that we followed the journal format, and it requires 5 to 7 figure in main articles.
- Thus, we were limited to include many figures in the main manuscript, and we chose to provide the all-important information supplementary materials.

- However, as per your suggestions, we have clarified the links between the paragraphs to further fluidity of the text.

"Line 161: While nano-DIHM determined and classified..." this statement ends with the assertion that it did not perform as well as high-resolution S/TEM; however, it's just not possible. DIHM cannot perform as well because of the different wavelength scales involved. This reads as if the authors expect that they could achieve the same resolution with a little more work. Be clear that it is not possible."

Response: Thank you very much. In the revised manuscript, we simplified the statement, and now it can read as:

“The nano-DIHM determined and classified the overall shape of virus particles, but it did not decipher the precise cluster shape of MS2 as compared to the high-resolution S/TEM. Nano-DIHM offers promising results for determining the phase, shape, size, and surface properties of airborne/waterborne particles. Currently, high-resolution electron microscopy is a powerful measurement tool for determining virus shape/morphology^{10,14,15}. But, electron microscopy does not have the in situ and real-time imaging capabilities^{10,14,15} that nano-DIHM offers¹ on the millisecond time scale.”

"Table 1: What are these images? Are these reconstructions or holograms? Some look like reconstructed images, while others have the look and feel of a hologram. Clarify".

Response: Thank you very much for your comment.

- Yes, these are the reconstructed images of objects/particles, and it shows the shape of particles.
- These objects/images are obtained by automated classification using Stingray software.
- We have clarified them in the revised manuscript in the Table 1 caption.

"Line 212: When discussing the size of the MS2 particles, be sure to make clear that there are aggregates of the viral particles."

Response: Thank you very much for your suggestions. In the revised manuscript, we double-checked and ensured consistency with our terminologies.

"Line 247: Figure 5 is cited out of order. Reference the figures in the order in which they appear."

Response: Thank you. We corrected the figures' order in the revised manuscript and ensured they were appropriately cited.

"Line 261: What is the depth of field of the objective?"

Response: Thank you for your question. The theoretical depth of field of the objective was 362 nm.

"Line 348: Quantify real-time. Regarding the classifier, It appears to be a binary (YES/NO) classification routine. What data is in the reference database? How many different types is the classifier comparing the know samples? What is the characteristic that most often triggers a successful classification? How long

is this whole process? How large is the database? What is large enough to achieve accurate classification?"

Response: Thank you very much for your comment.

- Stingray software is based on a patented algorithm^{8,9} that was trained to achieve the real-time in situ automatic detection, classification and physicochemical characterization of SARS-COV-2 by using nano-DIHM.
- The stingray software workflow follows three main categories: 1) Identify and find objects from recorded or real-time holograms, 2) classify the objects into taxon and 3) start training the classifiers. The Stingray software training procedure was given in methods. The basic algorithm of stingray software follows the Kirchhoff-Fresnel reconstruction approach, including robust deep neural network classifiers that extract in-focus objects and classify them within the image volume. The intensity threshold and edge gradient value will be used as input parameters to detect the virus or other objects.
- Indeed, the results are presented in YES/No format, which looks like a binary. However, the whole classification process is not binary, but our interest was to distinguish and classify only SARS-CoV-2 viral particles from the mixed sample of SARS-CoV-2 and MS2 bacteriophage. Hence, we presented the outcome results in a YES/NO format. By extracting the output results as a YES/NO format, we could use Nano-DIHM as an airborne or waterborne virus detector.
- Simultaneously, we obtained the physicochemical properties of the SARS-CoV-2 process, as presented in Table 1.
- We used seven different types of classifiers to compare the results. They are included: MS2 (dry aerosols), MS2 (moist droplets), TiO₂ (dry aerosols), 200 nm PSL (dry aerosols), SARS-CoV-2 + SARS-CoV-2 RNA + TiO₂ (water), SARS-CoV-2 + MS2 (air and water).
- The stingray software workflow follows three main categories: 1) Identify and find objects from recorded or real-time holograms, 2) classify the objects into taxon and 3) start training the classifiers. The basic algorithm of stingray software follows the Kirchhoff-Fresnel reconstruction approach, including robust deep neural network classifiers that extract in-focus objects and classify them within the image volume. The intensity threshold, edge gradient value, and shape information were input parameters to detect the virus or other objects. In earlier studies, the threshold intensity was a primary quantity to use for detecting and classifying objects^{2,16-18}.
- The complete detection, identification, and classification of viruses in real-time in situ of pairs of holograms take a maximum of 30 seconds.
- In each set of experiments, 10K holograms and 100k iterations were used to train the Stingray software, and 99% accuracy was achieved.
- Once the Stingray software trained as we achieved the ~ 99% accuracy was 10K hologram and 100k iteration. Further, 1-minute data is good enough for real-time testing to get 90 – 95 % accuracy results. This accuracy may be varied based on sample to sample.

We have provided comprehensive details of the classification and training of Stingray software in the method section from lines 545 - 581.

"Line 473: The authors state, "...nano-DIHM will be used..." Will it? This is a bold statement. Use the more conservative "may be used" or "can be used."

Response: Thank you. As per your suggestion, we replace it as can be used.

"Line 486: Mentions the setup, but there is no accompanying figure. Include a detailed apparatus schematic in the manuscript at the appropriate point."

Response: Thank you very much. The schematic of the experimental setup was given in Supplementary Figure S1. As per your suggestion, now, we moved Fig S1 into the revised manuscript, as Figure 7.

"Line 511: SMPS and OPS are, I believe, used as acronyms before they are defined."

Response: Thank you. We have included it, as suggested.

"Line 603: A more extensive..." How much more extensive? How many more samples? The experiments are somewhat limited in this regard, and the authors have not been clear in the manuscript."

Response: Thank you very much for your question.

- We have tested and experimented with the best possible scenario by mixing viral particles with metals, organics, and plastics to test the accuracy of the physicochemical detection of SARS-CoV-2 viruses.
- The Nano-DIHM provided successful real-time detection and physicochemical observation in situ.
- To clarify, we meant for more extensive library to identify or target unknown species. As in the real environment, several known or unknown species existed; without information on targeting species, Nano-DIHM cannot extract the information of unknown species.
- We have shown that Nano-DIHM successfully detected and classified the oil spills in water samples ².
- However, the next generation nano-DIHM may detect the unknown particles if they are viruses or not. Since even unknown viruses have physicochemical characteristics, we may have the rapid training of the software and confirmation with more conventional PCR techniques in future.

We would like to cordially thank both reviewers and the editor for their time, contributions, and comments. We truly appreciate them, and we trust that they are satisfied with our modifications that address all their concerns.

References used in this reply

- 1 Pal, D., Nazarenko, Y., Preston, T. C. & Ariya, P. A. Advancing the science of dynamic airborne nanosized particles using Nano-DIHM. *Communications Chemistry* **4**, 170, doi:10.1038/s42004-021-00609-9 (2021).
- 2 Hall, R., Pal, D. & Ariya, P. A. Novel Dynamic Technique, Nano-DIHM, for Rapid Detection of Oil, Heavy Metals, and Biological Spills in Aquatic Systems. *Analytical Chemistry* **94**, 11390-11400, doi:10.1021/acs.analchem.2c02396 (2022).
- 3 imaging, D. i. Desktop Microscope User Guide. (2018).
- 4 Garcia-Sucerquia, J. *et al.* Digital in-line holographic microscopy. *Appl. Opt.* **45**, 836-850 (2006).
- 5 Xu, W., Jericho, M. H., Meinertzhagen, I. A. & Kreuzer, H. J. Digital in-line holography of microspheres. *Appl. Opt.* **41**, 5367-5375, doi:10.1364/AO.41.005367 (2002).
- 6 imaging, D. i. Octopus Software User Guide. *Copyright © 4Deep inwater imaging 2018* (2018).
- 7 imaging, D. i. Stingray Software User Guide. *Copyright © 4Deep inwater imaging 2018* (2018).
- 8 Kreuzer, H. J. HOLOGRAPHIC MICROSCOPE AND METHOD OF HOLOGRAM RECONSTRUCTION. *US. Patent 6411406 B1, Canadian Patent CA 2376395* (2002).
- 9 Kreuzer, H. J. US. Patent 6411406 B1, Canadian Patent CA 2376395. (2002).
- 10 Pal, D., Dastoor, A. & Ariya, P. A. Aerosols in an urban cold climate: Physical and chemical characteristics of nanoparticles. *Urban Climate* **34**, 100713, doi:<https://doi.org/10.1016/j.uclim.2020.100713> (2020).
- 11 Rangel-Alvarado, R., Pal, D. & Ariya, P. PM2.5 decadal data in cold vs. mild climate airports: COVID-19 era and a call for sustainable air quality policy. *Environmental Science and Pollution Research* **29**, 58133-58148, doi:10.1007/s11356-022-19708-8 (2022).
- 12 Rahim, M., Pal, D. & Ariya, P. Physicochemical studies of aerosols at Montreal Trudeau Airport: The importance of airborne nanoparticles containing metal contaminants. *Environmental Pollution* **246**, doi:10.1016/j.envpol.2018.12.050 (2018).
- 13 Hinds, W. C. (Wiley & Sons New York, 1999).
- 14 Bhardwaj, J. *et al.* Recent advancements in the measurement of pathogenic airborne viruses. *Journal of Hazardous Materials* **420**, 126574, doi:<https://doi.org/10.1016/j.jhazmat.2021.126574> (2021).
- 15 Bhardwaj, J., Kim, M.-W. & Jang, J. Rapid Airborne Influenza Virus Quantification Using an Antibody-Based Electrochemical Paper Sensor and Electrostatic Particle Concentrator. *Environmental Science & Technology* **54**, 10700-10712, doi:10.1021/acs.est.0c00441 (2020).
- 16 Xu, W., Jericho, M., Meinertzhagen, I. & Kreuzer, H. Digital in-line holography for biological applications. *Proceedings of the National Academy of Sciences* **98**, 11301-11305 (2001).
- 17 Xu, W., Jericho, M. H., Kreuzer, H. J. & Meinertzhagen, I. A. Tracking particles in four dimensions with in-line holographic microscopy. *Optics Letters* **28**, 164-166, doi:10.1364/OL.28.000164 (2003).
- 18 Rotermund, L., Samson, J. & Kreuzer, H. J. J. M. S. R. D. A submersible holographic microscope for 4-D in-situ studies of micro-organisms in the ocean with intensity and quantitative phase imaging. **6**, 181 (2016).

REVIEWERS' COMMENTS:

Reviewer #2 (Remarks to the Author):

The authors have addressed my comments and questions.

Reviewer #3 (Remarks to the Author):

Four-dimensional in situ real-time physicochemical tracking of virus-laden droplets and aerosols in air

D. Pal, M. Amyot, C. Liang, P.A. Ariya

This manuscript reports on an apparatus that purports to perform real-time virus tracking using nano-digital in-line holographic microscopy. The authors claim to be able to characterize the time-dependent state of viruses and other particulates with respect to a variety of physicochemical properties.

A number of comments we made in the previous version of this manuscript. I note that the authors paid careful attention to the comments, and addressed all my concerns. The manuscript has been improved in content and clarity and has reached the point of acceptability for publication. At this point, the manuscript may proceed to publication.